# Challenges of the Green Transformation of Transport in Poland

Łukasz Brzeziński [1,*] and Adam Kolinski [2]

1 Faculty of Management and Logistics, Poznan School of Logistics, 60-755 Poznan, Poland
2 Lukasiewicz Research Network—Poznan Institute of Technology, 60-965 Poznan, Poland; adam.kolinski@pit.lukasiewicz.gov.pl
* Correspondence: lukasz.brzezinski@wsl.com.pl

**Abstract:** The transition to more eco-friendly forms of transport is one of the main challenges for the Polish economy in the coming decades. Poland, as a member of the European Union, must adapt to the requirements regarding, in particular, reducing carbon dioxide emissions related to new vehicles. The implementation of these changes will require significant financial outlays and structural reconstruction of transport (both public and private), as well as remodeling of the functioning and habits of society. The aim of this study was to identify and analyze the challenges of the green transformation of transport in Poland. The following research methods were used: desk research, focus interview–expert research, and an original approach to the use of SWOT analysis. Based on the conducted analyses, Poland's strategic position in the context of the "transport greening" process was determined. The strategy is in line with the concept of "reorganization", calling for a thorough restructuring of the development strategy. This entails coordinated efforts, such as conducting in-depth evaluations of current strategies, securing increased funding, providing support for research, and implementing public education initiatives. In essence, the study emphasizes the necessity for significant endeavors to effectively manage the green transition of transportation in Poland.

**Keywords:** green transformation; green transformation of transport; challenges of green transformation; strategy of green transformation; greening transport

## 1. Introduction

Low-emission transport is one of the key megatrends that, in the coming decades, will influence not only the way we travel but also our everyday lives. The transition to more eco-friendly forms of transport will become one of the main challenges for the Polish economy in the coming decades. Poland, as a member of the European Union, must adapt to the requirements for reducing carbon dioxide emissions in the case of new vehicles. The implementation of these changes will require significant financial outlays and taking into account social interests. A particular challenge will be to reduce emissions in the truck transport sector, which has become a key element of European road transport in recent years. In order to continue to compete in Western European markets, transport companies will be forced to invest in modern vehicle fleets.

In accordance with the guidelines of the government strategy of Poland's Energy Policy until 2040 [1], the transformation of our country's energy sector will require significant investments, the scale of which is estimated at approximately PLN 1.6 trillion in the period from 2021 to 2040. Of this amount, approximately PLN 867–890 billion may be invested in the fuel and energy sector, while in other sectors, such as industry, households, services, transport, and agriculture, this amount may reach approximately PLN 745 billion. Additionally, there is a real threat of an increase in the operating costs of vehicles with combustion engines, resulting, for example, from increased fuel taxation.

According to the EU document "2030 climate target plan: extension of European Emission Trading System (ETS) to transport emissions", it is likely that $CO_2$ emissions from the transport sector will be covered by the European Emissions Trading System

(ETS) as early as 2026. The direct changes will mainly affect fuel distributors, not car owners, but the expected effect will be an increase in fuel prices. As part of the energy transformation in Poland, it is planned to use national competitive advantages, which are to lead to new development opportunities and the initiation of broad modernization changes. It is estimated that such initiatives can create up to 300,000 new jobs, especially in sectors related to renewable energy, nuclear energy, electromobility, network infrastructure, digitalization, thermal modernization of buildings, etc.

Additionally, the current vision of the green transformation of transport in a sustainable way is as follows: the transport of the future will be based on the synergy of various technologies. In addition to increasingly advanced electric motors, which will be powered by more and more efficient batteries, hydrogen fuel cells, next-generation biofuels, synthetic fuels, and gas will also be used. Nevertheless, these assumptions have certain limitations. They are based on current knowledge and experience, while thousands of scientists and engineers are constantly working on even more innovative solutions. The effects of this work will only be known in the future, and the developed technologies will be implemented in a few or a dozen years. According to current forecasts, technical solutions that are not yet commercially available will be responsible for almost half of the global reduction in $CO_2$ emissions by 2050. It is obvious that it is necessary to support research and development and provide commercial support for each technology that will contribute to achieving the established climate goals [2]. Such costly and demanding social changes, which are to remodel the functioning of many sectors of the economy, should be implemented based on the adopted plan as well as a detailed analysis of possible opportunities, barriers, and the internal situation of the country.

In this article, as part of the desk research, a bibliometric analysis was also carried out relating to publications that are thematically related to the green transformation of transport, thanks to which the conducted research in this field was systematized from an international perspective, showing trends and research directions. This provided a comprehensive overview of current research and developments. This identified areas where there is a lack of research or insufficient data. One of these areas (research gaps) is determining the validity, possibilities, or potential for the green transformation of transport at the national (or regional) level, with a focus on developing strategic assumptions for such transformations. Therefore, the authors undertook research in this area. The added value of the study relates both to the systematization of the current state of knowledge as well as to the determination of practical premises for the implementation of a project of such a significant scale for society and the economy.

The aim of the article is to identify and analyze the challenges of the green transformation of transport in Poland. The researchers' leading methods are expert interviews and SWOT analysis (modified by the author to suit the specifics of the article).

The following research problem was formulated:

- Q1: What are Poland's strengths in the context of the green transformation of transport (based on expert opinions)?
- Q2: What are Poland's weaknesses in the context of the green transformation of transport (based on expert opinions)?
- Q3: What opportunities can the green transformation of transport bring for Poland (based on expert opinions)?
- Q4: What threats does the green transformation of transport generate for Poland (based on expert opinions)?

The article consists of six parts: an introduction, a review of the literature on the subject, a description of the methodological assumptions of the author's research, an analysis of the research results, a scientific discussion of the results, and a summary.

## 2. Literature Review

The section describes issues related to defining the essence of the green transformation of transport, the conditions of this process, determining the course in Poland, and the solutions and technologies used.

### 2.1. Green Transition: Concept Analysis

The green transformation is an important area of research undertaken by authors from various fields of science. These publications cover both definitional issues of the concept of a "green transformation" and contemporary analyses regarding the current progress of the transformation process or the implementation of specific solutions. As Bąk and Cheba point out [3], the terms "green transformation" and "green transition" appear equally often in the literature on the subject. Some authors treat them interchangeably [4]. However, there are approaches according to which they are differentiated. The concept of green transformation refers to much more complex processes in the political, social, and technological spheres [5]. In turn, the term "transition" describes an orderly process carried out according to precisely defined technical knowledge [6].

It should be noted that both of these terms are defined in various ways in the literature on the subject. The research part defines the green transition as a transition towards achieving climate and environmental goals mainly through investments [7]. The (green) transformation itself is associated with the environmental effects (often benefits) of changes and the protection of natural resources that are intended to counteract unfavorable climate changes [8,9]. On this basis, it can be concluded that the green transformation should be understood as the restructuring of political, social, and economic systems in order to preserve or improve the condition of the planet for future generations. A similar approach is presented by Newell [10], emphasizing that industrial societies should strive for transformation towards a climate-compatible, resource-saving and sustainable global economic order.

In addition, when defining these concepts, some authors emphasize certain aspects related to the green transformation, such as security issues (e.g., energy) [11], just transformation, i.e., technologies and solutions available to various social groups [12], or focus on a low-emission or even zero-emission economy [13]. The green transformation is also associated with technological progress towards greening the economy [14].

It can therefore be indicated that green transformation is a concept that encompasses the transition towards a more sustainable and environmentally friendly society. This process involves making changes in various sectors such as energy, transportation, agriculture, and manufacturing to reduce environmental impacts and promote the efficient use of resources.

### 2.2. Green Transformation Background

Initially, global efforts concentrated on minimizing specific detrimental emissions, but as time progressed, the broader aspects of sustainable development, encompassing social, economic, and ecological dimensions, came into focus. Additionally, there has been a growing acknowledgment of the interconnectedness and the necessity to harmonize economic advancement, societal welfare, and environmental preservation, marking them as paramount priorities [15].

Formulating an international agreement regarding actions fostering the green transition grounded in socio-economic and environmental principles is a protracted endeavor. It is important to recognize that the green transition entails a gradual move toward economically sustainable development and an economy reliant on low-carbon alternatives, as opposed to fossil fuels and the overconsumption of natural resources.

Issues of environmental protection and sustainable development already have a certain history and constitute an important element of the functioning of countries in the international environment.

In this field, the United Nations Environmental Program can be particularly distinguished. Its goal is to coordinate UN activities in the field of environmental protection

and constantly monitor its condition around the world. These activities are undertaken based on the concept of sustainable development. Initiatives undertaken by UNEP concern both nature protection (e.g., protection of biodiversity, desert areas, waters, etc.) and take into account environmental requirements in the economy, e.g., in construction, tourism, investments (e.g., the promotion of renewable energy sources), etc. [16].

The program participates in almost all global initiatives in the field of environmental monitoring. These activities are carried out in cooperation with other international organizations, including the World Meteorological Organization and the Global International Waters Assessment. One of the manifestations of UNEP's activities is the development of international environmental protection law. This institution contributed to the creation of such agreements as the 1973 Convention on International Trade in Endangered Species of Wild Fauna and Flora, the 1979 Bonn Convention on the Protection of Migratory Species of Wild Animals, the 1985 Vienna Convention for the Protection of the Ozone Layer, the Montreal Protocol on Substances that Deplete the Ozone Layer 1987, the Basel Convention on the International Transport of Hazardous Wastes 1989, the United Nations Framework Convention on Climate Change 1992, the United Nations Framework Convention on the Conservation of Biological Diversity 1992, the 1994 Convention to Combat Desertification, and the 2001 Stockholm Convention on Persistent Organic Pollutants [17].

However, the inaugural global environmental endeavor emerged with the signing of the Montreal Protocol on 16 September 1987 [18]. Its primary aim was to safeguard the ozone layer, yet it fell short of expectations due to various deficiencies, notably its non-binding nature for signatories. Moreover, the agreement failed to foster sustainable development or tailor measures to individual national circumstances.

In a groundbreaking move, the international community, during the 1992 United Nations Framework Convention on Climate Change (UNFCCC) [19], established the objective of curtailing greenhouse gas (GHG) emissions. Shortly after the UNFCCC came into effect [20], negotiations commenced in 1994, culminating in the adoption of the multilateral Kyoto Protocol in December 1997 [21].

The Kyoto Protocol aimed to slash the GHG emissions of 37 industrialized nations and the European Union (EU) during its initial 2008–2012 implementation phase. Developed nations pledged to reduce GHG emissions by 5% relative to 1990 levels, while EU member states committed to an 8% reduction [22]. Unlike the Montreal Protocol, the Kyoto Protocol was legally binding and underscored the pursuit of sustainable development objectives, particularly emphasizing energy efficiency, sustainable agriculture, and the formulation of tailored national measures [23].

A significant stride was the establishment of mechanisms for joint protocol implementation (International Emissions Trading, Joint Implementation, and the Clean Development Mechanism), which laid the groundwork for further collaboration and action between developed and developing nations [24].

Despite the majority of countries ratifying the Kyoto Protocol in 2005, it fell short of expectations and failed to establish environmental stability worldwide. The principal reasons were the United States, the largest emitter, refusing to ratify it and Canada withdrawing from the Protocol in 2011. For years, most countries failed to implement concrete measures, while developing nations notably increased their emissions, exacerbating global greenhouse gas (GHG) levels. In December 2012, during the Doha Conference in Qatar, a second commitment period was initiated under the Kyoto Protocol, known as the Doha Amendment. Signatories pledged to reduce GHG emissions by at least 18% from 1990 levels between 2013 and 2020. Over the 15 years since the Kyoto Protocol's inception, the persistent rise in GHG emissions has prompted renewed public inquiry into consensus-building for addressing global warming [25].

The Paris Agreement, ratified on 12 December 2015, during the 21st Conference of the Parties to the UNFCCC, aims to mobilize global efforts to combat climate change beyond 2020 [26]. As the first legally binding climate pact applicable to all nations, the Paris Agreement outlines strategies for achieving climate neutrality (such as afforestation, investments

in renewable energy, and implementing a carbon tax on imports from non-compliant countries). Moreover, it delineates provisions for financial and technical assistance to developing nations, facilitates technology transfer, and supports capacity-building efforts.

In contrast to the Kyoto Protocol's requirement for developed nations to cut gas emissions, the Paris Agreement mandates the involvement of all countries worldwide through nationally determined contributions (NDCs) and the implementation of domestic measures to achieve set objectives, with regular reporting on progress. Countries must periodically update their NDCs every 5 years, each successive one being more ambitious than the last, to meet and progressively elevate their targets. Among nations, the European Union (EU) has made significant strides, propelled by political decisions and reforms outlined in the European Green Deal [27], a strategic blueprint guiding future economic development.

The EU vigorously advocates the green transition as a prolonged endeavor aimed at transforming and decarbonizing the economy and fostering well-being through a sustainable economic model that integrates socio-economic and ecological dimensions of development [28]. This transition encompasses all sectors of the economy beyond energy, embracing business models conducive to decarbonization and adhering to circular economy principles. Consequently, the EU systematically supports green innovations, technologies, and investments across its public policies [29], extending such expectations to candidate countries as well.

The European Commission unveiled the Roadmap to the Green Deal [30] in December 2019, outlining a comprehensive long-term development strategy [31] aimed at positioning Europe as the world's first climate-neutral continent by 2050, with a targeted emission reduction of 55% compared to 1990 levels. While all EU member states are individually committed to achieving climate neutrality, five of them have enshrined this goal into law: Sweden targeting 2045, and Denmark, France, Germany, and Hungary aiming for 2050.

Central to the Green Deal's agenda is the transition to clean energy and sustainable resource management, fostering avenues for innovation, investment, and job creation. The envisioned benefits of the Green Deal encompass improved air quality, access to clean water, the preservation of soil health and biodiversity, the renovation of energy-efficient buildings, affordable and nutritious food options, expanded public transportation networks, the adoption of cleaner energy sources, and the advancement of cutting-edge clean technologies. Additionally, the initiative seeks to promote the production of durable goods that can be repaired, recycled, and reused, while also fostering the development of future-proof jobs and providing skills training for the transition, thus bolstering global competitiveness and industrial resilience [28,32].

The EU has enshrined climate neutrality within its regulatory framework, exemplified by the adoption of the inaugural European Climate Law in 2021 [33], which comprises a suite of 55 regulations, notably including the following:

- The revised Renewable Energy Directive, which increases the obligation to participate in the production of renewable energy sources by 8% by 2030;
- The revised Energy Efficiency Directive, which introduces a public sector obligation to renovate 3% of publicly owned buildings each year;
- The revised Energy Taxation Directive, which introduces new forms of taxation of energy products in line with climate goals;
- New regulations to promote higher standards for car and van emissions;
- The revised Alternative Fuels Infrastructure Regulation, which includes the installation of infrastructure for electric charging and fuel;
- A new set of regulations as a guideline for land, forest, and agricultural use towards achieving EU carbon removal targets.

Furthermore, all EU member states have unanimously agreed to phase out all direct or indirect subsidies for fossil fuels by 2025 [34]. The updated regulatory framework for the transportation sector envisions the inclusion of road traffic in emissions trading after 2026 while concurrently incentivizing the utilization of renewable energy sources and investments in novel clean technologies through subsidies. Additionally, for the aviation

industry, proposals include the implementation of a pollution tax and the promotion of sustainable aviation fuels, mandating the adoption of sustainable blended fuels for all departures from EU airports. In maritime transportation, the extension of carbon pricing to this sector and the reduction in pollutant fuel usage that harms local environments are proposed.

At the forefront of the fourth industrial revolution [35], the green transition presents European industry with the opportunity to cultivate markets for clean technologies and products, influencing value chains across the energy, transportation, and construction sectors. The transition towards a sustainable economy through electrification and increased reliance on renewable energy sources holds potential for heightened employment rates within these sectors. The drive to enhance energy efficiency in buildings is poised to generate job opportunities in construction, fostering a demand for local labor.

Given that numerous EU enterprises engage in importing goods from countries and regions beyond the EU, circumstances conducive to unfair competition arise. It is stipulated that importing firms are obligated to cover the carbon price, prompting the establishment of a specialized Cross Border Adjustment Mechanism (CBAM) [36]. The CBAM constitutes an additional levy levied by the EU on imports of carbon-intensive commodities (such as iron and steel, aluminum, cement, fertilizers, electricity, and hydrogen) from non-EU territories, aimed at thwarting carbon leakage (the relocation of production to regions with less stringent climate policies and subsequent importation of these goods into the EU) [37]. The implementation of this tax directly inflates product costs, thereby adversely impacting the price competitiveness of the designated goods and rerouting consumption away from nations that lack carbon emission taxation.

There is a broad consensus that adopting renewable fuels will significantly decrease energy consumption, emissions, and energy expenses for both consumers and industries. Equally crucial is the alignment of the energy tax system with the green transition, ensuring minimal tax rates to aid vulnerable citizens. To address the challenges posed by the energy transition, particularly in regions heavily reliant on energy- and carbon-intensive industries or fossil fuel-powered electricity systems, the EU has established a new Just Transition Fund.

The Effort Sharing Regulation represents a strategic initiative aimed at establishing national targets for reducing greenhouse gas (GHG) emissions, aligning with the EU's commitments under the Paris Agreement. Targeting sectors responsible for over 60% of total emissions, namely transport, agriculture, buildings, and waste management, the objective is to achieve a 30% reduction in emissions by 2030 compared to 2005 levels [37]. To ensure member states' active participation in emission reduction efforts within these sectors, the regulation imposes minimum binding annual GHG emission targets across EU countries. Given the varying capacities among member states to reduce emissions [38,39], targets are determined based on the gross domestic product (GDP). A safety margin of 105 million tons of $CO_2$ equivalent is projected to be available by 2032, intended particularly to assist less affluent EU member states in meeting their 2030 targets. While access to this reserve is contingent upon the EU reaching its 2030 goal under stringent conditions, some flexibility may be permitted if EU countries opt to borrow and transfer annual emission allocations from one year to the next.

Despite the EU's dedicated commitment to the decarbonization agenda, the concept of resilience has emerged as a pivotal point for EU policies. Resilience is deemed essential for the EU and its individual member states to confront and adapt to global challenges while navigating the transition in a sustainable, equitable, and democratic manner [40]. The repercussions of the COVID-19 pandemic and the energy crisis underscored various challenges not only for the EU economy [41], marked by unstable energy provision, fluctuating food and resource costs, and disruptions in supply chains [42], but also for society at large, with vulnerable demographics bearing the brunt.

Initiatives such as the 2020 Strategic Foresight Report [43] and A Strategic Compass for Security and Defence [44] exemplify Europe's endeavors to bolster its resilience, particularly concerning climate, defense, and energy, shaping responses within the realms of green and digital transitions. Recognizing that resilience demands adaptability and swift responsiveness and considering the intricate and enduring nature of the green transition, it becomes apparent that, particularly post-2022, the EU grapples with gaps in critical areas like energy, food, and resources in the absence of comprehensive data concerning the scale and ramifications of these challenges.

It is worth noting that the driving axis of changes and the implementation of environmentally friendly technologies are, in particular, political conditions, which are particularly strong in the European Union. However, the described phenomena are not a decisive factor that accelerates eco-friendly transformation [45]. It is possible to point to other phenomena occurring both in social awareness and in the economy. An example may be the increase in the price of energy carriers as a result of the armed conflict in Ukraine. For this reason, the use of electric vehicles becomes much less commercially attractive from the users' point of view. In addition, one can point to research by Hess [46] that shows that, in industrialized countries, there is tension between the need to switch to low-emission energy sources and the desire to meet the rapid increase in energy demand while continuing to exploit fossil fuels. Also interesting are the studies presented by Ouyang et al. [47]. The indicated authors used a two-way panel model with fixed effects to analyze the impact of environmental regulations on technological innovation. The study showed a U-shaped relationship between these variables. In the short term, environmental regulations have a negative impact on the development of research and innovation capabilities in the industrial sector in China. And with expanding environmental regulations, organizations are trying to reduce pollution control costs by improving their technological innovation capabilities.

*2.3. Green Transformation of Transport: Systematic Literature Review*

The green transformation of transport refers to the process of restructuring transport systems in a way that minimizes negative impacts on the environment and promotes sustainable solutions. This includes changes in transport infrastructure, modes of transport, transport policy, and travel habits. It is a long-term and multidimensional process.

Over the last two decades, this phenomenon has been considered in many different publications. Based on a review of the literature available in the Scopus database, over the period 2000–2023, there were 446 indexed publications that referred in their title, abstract, and/or keywords to the term "green transformation transportation".

There has been a noticeable systematic increase in publications on the analyzed topics. In individual years, the number of publications fluctuated to some extent, but a dynamic increase was recorded in the years 2015–2023 (an increase from 5 to 114 publications per year). A similar tendency occurred in the number of citations; in fact, there was a systematic increase throughout the analyzed period. In 2015, the number of citations (per year) was 101. In the following years, there was a dynamic increase to 1263 in 2023 (Figure 1).

It is interesting that, despite the green transformation (including transport) being perceived as European region-led, the largest share in the number of publications on the indicated topics were from China (215), the USA (44), India (21), Germany (16), Italy (16), Canada (12), Australia (11), Poland (11), and Turkey (11) (Figure 2).

It should be noted that aspects relating to the green transformation of transport are interdisciplinary. Considering the subject area of the analyzed publications, we can distinguish, in particular, engineering (17.6%), environmental science (15.0%), social sciences (11.4%), and energy (11.0%) (Figure 3).

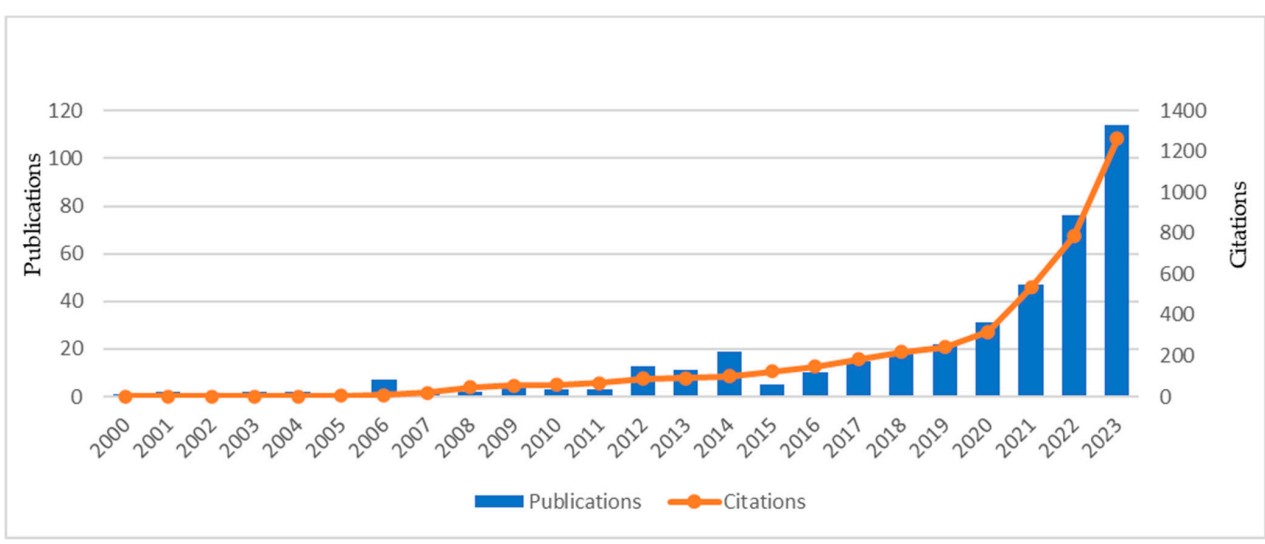

**Figure 1.** Number of publications and citations indexed in the Scopus database, including indicated keywords. Source: own study.

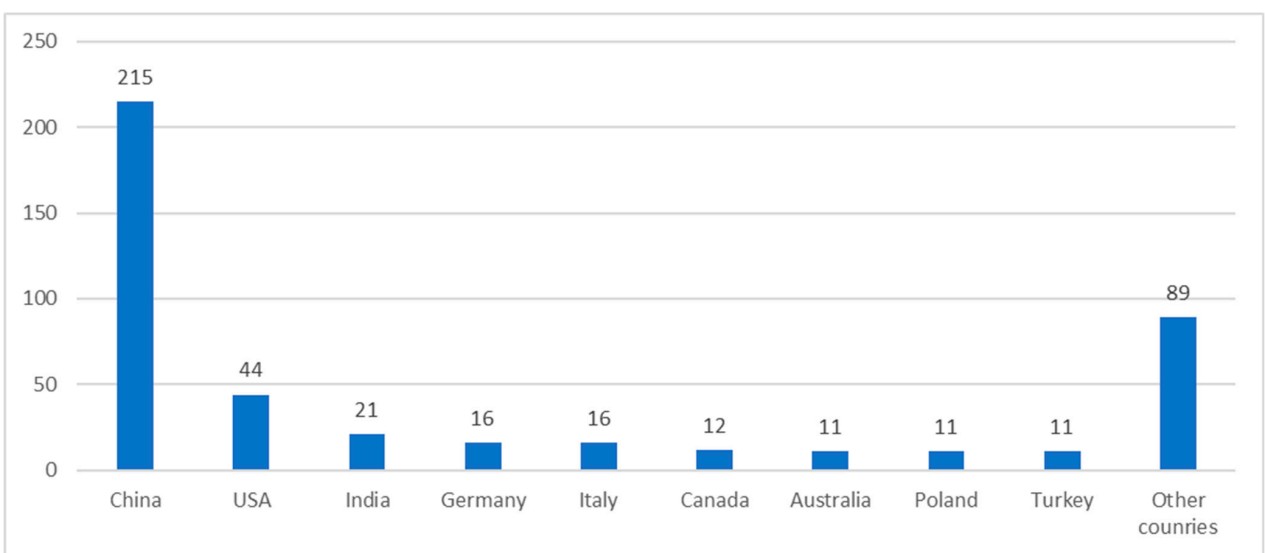

**Figure 2.** Publications by country. Source: own study.

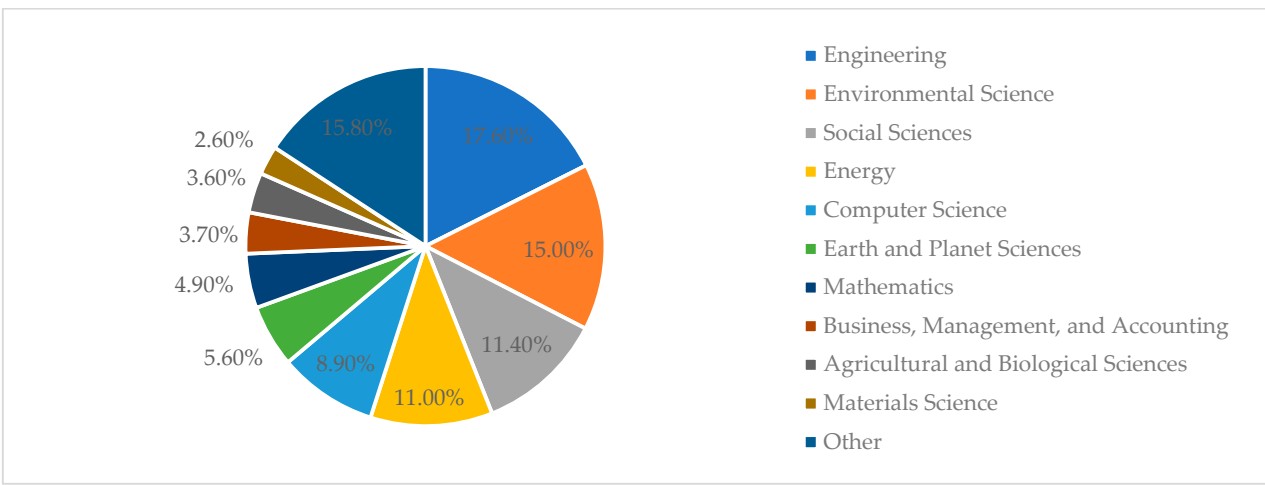

**Figure 3.** Publications by subject area. Source: own study.

The specificity of the green transformation of transport is the transition from traditional, often polluting, and environmentally unfavorable forms of transport to more eco-friendly and sustainable alternatives. This includes a range of activities that change the way society travels and moves and the infrastructure that supports this new transport model. Here are some key features of the green transport transformation:

- Shifting modes of transport: shifting from individual combustion vehicles to greener options, such as electric vehicles, hybrid vehicles, electric trains, and public transport. Promoting walking, cycling, and alternative forms of transport, such as carsharing and ridesharing, is also an important element.
- Development of environmentally friendly infrastructure: investments in infrastructure supporting sustainable transport, including a network of bicycle routes, sidewalks, public transport systems, and charging stations for electric vehicles. Striving to create more compact, green cities with a greater share of space for pedestrians and cyclists.
- Education and changing travel habits: promoting environmental awareness among the public and education about the benefits of green transport. Encouraging people to change their travel habits by limiting individual driving in favor of public transport, cycling, or walking.
- Use of new technologies: introducing innovative technological solutions, such as road traffic management systems based on artificial intelligence, applications supporting public transport, and intelligent shared transport systems.
- Policy and regulations: creating and implementing policies and regulations supporting the green transformation of transport, including charges for the use of combustion vehicles in city centers, subsidies for electric vehicles, investments in public transport infrastructure, and financial incentives for enterprises and individuals using green transport solutions.
- Sustainable spatial planning: integrating the green transformation of transport with spatial planning, ensuring access to public transport, services, and jobs in a sustainable way that is accessible to all residents.

It should be stated that the specificity of the green transformation of transport consists of a comprehensive approach that takes into account various economic, social, and environmental aspects, aiming to create a more sustainable and environmentally friendly transport system.

### 2.4. The Path to Greening Transport in Poland

Greening transport will be one of the main challenges for the Polish economy in the next few decades. As a member of the European Union, it must comply with the carbon dioxide emission reduction target for new cars. The changes will be extremely costly and must take into account the interests of the entire society. Reducing emissions in truck transport will be a particular challenge. In recent years, Poland has become a European power in road transport. If they want to operate in the Western European market, transport companies will be forced to invest in modern rolling stock. Attempts to postpone changes in just a few years may lead to a deterioration in the competitiveness of the economy and a decline in the attractiveness of Poland as a country to invest in or visit. At the same time, the green revolution in transport creates an important opportunity to become fully independent from energy resources from Russia. Transport accounts for roughly a quarter of the world's carbon dioxide emissions. In the European Union and Poland, the share of emissions from transport is currently higher compared to 1990, unlike, for example, industry, whose emissions have been successfully reduced. In 2020, only just over 10% of transport in the EU used green energy. Already in 2030, Brussels wants to increase this share to 14% [2].

Poland ranks eighth in the European Union in terms of $CO_2$ emissions from the transport and logistics sector. However, the share of emissions from transport is proportionally smaller than the position of our entire economy (Poland is sixth in the EU). In 2019, the share of emissions from the transport sector of Poland was 4.76% of the total emissions from

transport and logistics in the EU. It is worth noting that, compared to Poland, countries with a developed maritime economy have a greater share of $CO_2$ emissions from transport. On the other hand, the volume of $CO_2$ emissions from transport in Poland has been growing faster than the European average in recent years. Pre-COVID-19, in 2019, it was as much as 27% higher than in 2011. In the same period in the European Union, transport emissions increased by less than 11% [2].

More than 25 years ago, the European Union attempted to reduce $CO_2$ emissions from the road transport sector. In 1995, the European Commission adopted a strategy for reducing $CO_2$ emissions from passenger cars based on three pillars: voluntary agreement with representatives of the automotive industry, improving access to information for consumers, and promoting fuel efficiency through fiscal measures. At the end of 2019, the European Green Deal was announced. It envisages achieving "climate neutrality" by the European Union by 2050. Neutrality is a state in which the $CO_2$ emission balance is zero, i.e., the emitted carbon dioxide is entirely absorbed by plants or captured by special industrial installations.

When discussing the importance of Fit for 55 for transport, several elements should be indicated [48]:

- From 2035, the sale of passenger cars and light commercial vehicles with combustion engines in the EU will be subject to additional taxes to such an extent that the production of such vehicles and their introduction to the EU market will become practically unprofitable.
- By 1 January 2024, each member state was to submit to the European Commission a draft of a national strategy for the development of the alternative fuels market, ensuring the achievement of the objectives of building the Alternative Fuels Infrastructure Regulation (AFIR) infrastructure and the Fit for 55 package.
- At the end of each year, member states will ensure that a minimum of 1 kW of power is available in public charging infrastructure for each newly registered electric vehicle. For each PHEV vehicle, there should be a minimum of 0.66 kW of power.
- Operators of electric vehicle charging infrastructure will be obliged to provide clear information on the charging price, making it possible to obtain it before charging. This applies especially to the price per session, price per minute, and price per kWh.
- There should be clear and recognizable signage for charging points available within the Trans-European Transport Network (TEN-T), ensuring easy access to these points.
- By 31 December 2030, hydrogen refueling stations should be located every 150 km within the TEN-T network.
- Prices for fuel and power infrastructure should be "market-based and reasonable".

The energy transformation of transport requires a holistic approach and a coherent strategy. The optimal technology and way to move away from fossil fuels have not yet been invented. The experience of highly developed countries indicates the advisability of using various tools, including legal, fiscal, and educational ones. It will also be necessary to change communication habits and even lifestyles. The revolution in transport will not succeed without support and active state policies, both at the central and local levels. It cannot also take place at the expense of social tensions or the communication exclusion of certain social groups [49–53].

The goal of the green transformation of transport for the European Union is not to limit mobility but to change its philosophy and model. The volume of transported goods and the number of passengers will increase from year to year. Therefore, the EU transport policy is focused on the following goals [54]:

- Increasing the energy efficiency of vehicles, developing environmentally friendly fuels and drives;
- Optimization of multimodal supply chains, promotion and development of more effective freight transport solutions;
- Better use of existing transport infrastructure, including through transport management systems and the construction of an integrated European railway network.

### 2.4.1. Changes in Law

In 2017, the Polish government adopted two documents supporting the green transformation of transport: the "Electromobility Development Plan" (PRE) and the "National Policy Framework for the Development of Alternative Fuels Infrastructure". These documents specify the goals that the Polish market for vehicles with alternative drives should achieve by 2020 (in the largest agglomerations) and by 2025, throughout the country. These assumptions were changed in the Sustainable Transport Development Strategy to 2030.

As part of PRE, 50,000 electric vehicles and 3000 CNG-powered vehicles were to travel on Polish roads in 2020. By the end of 2025, the fleet of electric vehicles registered in Poland will number 1 million, and the fleet of CNG and LNG vehicles will number 54,000 and 3000, respectively. An ambitious goal was set for some of the supreme and central state administration bodies and the entities serving them. From 2025, their fleets should include at least 50% electric vehicles (BEVs).

In 2018, the Act on Electromobility and Alternative Fuels was adopted. Its aim is to stimulate demand for vehicles with alternative drives. The act introduced changes to the legal system that increase the attractiveness of using low-emission vehicles [2]:

- Possibility of use of bus lanes by electric vehicles (BEVs) until 1 January 2026;
- Free parking in paid parking zones for electric vehicles (BEVs) and unlimited entry to clean transport zones for fully electric vehicles powered by CNG or LNG;
- Indefinite exemption from excise tax on fully electric (BEV) and hydrogen (FCEV) passenger cars;
- Periodic exemption for plug-in hybrid cars, increased depreciation write-offs for wear and tear of passenger cars (BEV): up to PLN 30,000.

In 2021, the Council of Ministers adopted the Polish Energy Policy, to be effective until 2040. This document contains changed goals for electromobility. The strategy assumes that, in 2030, there should be 600,000 electric and hybrid vehicles registered in Poland. To ensure the possibility of charging, there should be 49,000 charging points with normal power and 11,000 points with high charging power in public charging stations by 2030. In a very ambitious variant, there should be 85,000 and 15,000 such points, respectively [55].

### 2.4.2. Support for Alternative Drives

Vehicles with alternative drives are still much more expensive than combustion cars. In order to encourage drivers to purchase them, state governments are introducing, among other promotions, subsidy systems. Such solutions were chosen in the largest European Union countries, including Germany, France, and Italy. Subsidies for BEV cars and FCEV cars amounted—depending on the country—from PLN 7000 to PLN 9000, and in the case of scrapping a previously owned car, the total incentive was up to PLN 12,000.

In turn, in the USA, a relief of USD 7500 was introduced for buyers of electric cars. However, it expires after a given brand reaches total sales of 200,000 electric vehicles. In addition to the federal incentive, buyers in the United States can also benefit from state-level relief and subsidies. Support for buyers of BEV and FCEV cars also exists in Poland. In mid-2020, recruitment for subsidy programs for the purchase of BEV passenger cars ("Green Car" and "Koliber") and delivery vans ("eVan") was launched. The subsidies were available to individual customers as well as companies and institutions. The subsidy for an "electric" passenger car under the "Green car" for individuals was up to PLN 18,750. However, the catalog of vehicles for which you could receive support was limited to models whose purchase price did not exceed PLN 125,000. An increase in the depreciation rate threshold is also an incentive for companies to invest in electric and hydrogen vehicles. In the case of electric and hydrogen cars, entrepreneurs can fully deduct expenses up to PLN 225,000 (in the case of others, the maximum depreciation threshold is PLN 150,000) [2].

### 2.4.3. Access to Charging Infrastructure

In addition to the price of vehicles and the cost of alternative fuels, the availability and friendliness of the refueling and charging infrastructure will determine the choice of

vehicles with alternative drives. At the end of 2020, the number of publicly available electric car charging stations in Poland, according to the European Alternative Fuels Observatory, was almost 1.7 thousand, which put the country only in 12th place among the European Union countries, including countries that are much smaller [1]. Although, in August 2022, there were over 4.4 thousand such stations in Poland, it was still far too few compared to the European leaders. Without a significant expansion of the electric charging infrastructure, the attractiveness of EV cars will remain limited. At the beginning of 2022, the "Support for electric vehicle charging infrastructure and hydrogen refueling infrastructure" program was implemented. It aims to support the development of infrastructure for charging electric vehicles and hydrogen refueling infrastructure, including introducing facilitations in the construction of charging points in multi-family buildings defining concepts necessary for the creation of hydrogen refueling infrastructure. In fact, entities interested in such installations (investors, housing communities) bear the investment costs as well as the associated risks [2].

### 2.4.4. Supporting Public Transport and Other Forms of Mobility

$CO_2$ emissions from transport in cities account for approximately 25% of greenhouse gas emissions from the transport sector. Therefore, limiting the use of combustion cars in cities creates the potential to reduce emissions. The way to reduce emissions is to use public transport more widely and support other forms of micromobility, such as city bike systems, carsharing, e-scooters, and even walking. In the future, a passenger car will serve as the so-called last mile, allowing you to travel from home to the transfer point. Since many people use their own cars to commute to work, school, shopping, or for other purposes, further investments are necessary in the expansion of agglomeration railways, additional rolling stock, and transfer hubs connected to park-and-ride parking lots. A separate issue of reducing pollutant emissions is not traveling. This was evident during the pandemic, when mobility restrictions significantly reduced fuel demand and therefore reduced emissions. The tools and models developed during this time, used for remote learning and work, may, in the long term, contribute to changing social habits in transport [56–58].

### 2.4.5. Clean Transport Zones

The Electromobility Act allows municipalities to introduce restrictions on car traffic in city centers in the form of so-called clean transport zones (SCT). Entry to them is only allowed for electric, hydrogen, and natural gas vehicles. The commune council may also allow entry free of charge to combustion vehicles specified in the resolution, e.g., those that meet specific emission standards. According to the amendment to the Act on Electromobility, clean transport zones will be able to be created in all municipalities, and the rules of their operation will be determined by the municipal authorities. There are currently about 250 cities in Europe with such zones. Poland's commitment to introduce green transport zones intended for zero- and low-emission cars has also been included in the National Reconstruction Plan. Brussels wants such zones to be introduced in cities with over 100,000 inhabitants, where air quality thresholds will be exceeded from the first quarter of 2025 [2].

### 2.4.6. Changes in Truck Transport

In addition to new types of truck drives, reducing the emissions of this category of vehicles will require a change in the formula of transport itself, e.g., an increase in the share of intermodal transport. Instead of transporting one load over long distances, trucks will more often travel shorter distances between intermodal terminals. This will reduce, for example, the number of empty courses and thus increase efficiency. According to the Transport and Logistics Poland Employers' Association (TLP), emission reduction is also possible thanks to the revision of EU law, which currently limits the efficiency of the use of means of transport in the transport of goods. One of the reasons for this state of affairs is the requirement introduced by the mobility package to return motor vehicles to the

operating base within at least eight weeks. The research company Ricardo estimates that this order could generate 2.9 million tons of additional $CO_2$ emissions in 2023 [59]. This means an increase in emissions from road transport in Europe of 4.6%. According to TLP, the real challenge is changes in the EU rules and conditions for road transport, limiting the so-called blank runs. Another solution to reduce emissions may be allowing the use of LHV vehicles on the TEN-T network, as well as creating conditions for the implementation and use of the so-called car platooning technology [2].

### 2.4.7. Changes in Social Attitudes

People's attitudes towards and use of transport are changing. This can be seen in Poland, although on a much smaller scale than in some Western European countries. Changes in habits will mainly concern travel within cities as well as to and from cities. Public transport will become more popular. In the case of Poland, changing the transport model also requires increasing social acceptance. Despite the improvement in the condition of public and agglomeration transport, many people still do not want to use this type of transport, pointing out, among others, its unpunctuality. The task of the International Energy Agency (IEA) is to invoke changes in society's behavior and approach to transport, which may result in a reduction in energy consumption by 10–15% by 2050 compared to 2020 [60]. The EU is discussing, among other initiatives, reducing the speed of cars on roads to 100 km/h. This would allow for a $CO_2$ reduction of 3% in 2030 compared to the current state.

The green revolution in transport is a challenge for hydrocarbon producers and suppliers. They will certainly do everything not to lose the market. They will be forced to find their way in a new legal and fiscal order where consumers are strongly discouraged from using their traditional products. It is extremely important that regulations aimed at greening transport are developed in the spirit of technological neutrality, ensuring support for the widest possible range of technical solutions and alternatives to fossil fuels. This will give entrepreneurs greater room to maneuver in moving away from their current activities based on hydrocarbons, and consumers will be able to choose solutions that meet their needs [61–65].

The transport industry, which is the lifeblood of the economy, will also face challenges. By remaining outside the current of changes visible in Western Europe, it will be doomed to failure, which will be felt by the entire Polish economy. This cannot be ignored. At the same time, the necessary changes must be implemented in a way that takes into account the interests of broad social groups and the level of wealth of Poles. The success of low- and zero-emission transport will depend not only on the availability and quality of technologies but also on their effectiveness and costs. However, the final effect of greening transport will probably be determined by a combination of the above-mentioned factors and everyday usability.

### 2.5. Technologies and Solutions for Green Transformation of Transport

The green transformation of transport is extremely universal. As presented by Hoppe and Trachsel [66], from a broader point of view, one can distinguish the following: vehicle technologies, engine technologies, material technologies, infrastructure and operational technologies, and emerging new mobility products and services.

### 2.5.1. Vehicle Technologies

Autonomous driving systems have emerged as a significant trend in the realm of vehicle technologies, showing substantial growth in recent years. While cars have been progressively equipped with various driving assistance features, such as cruise control, collision warning, and lane-keeping support, they are steadily moving towards full autonomy. The advent of autonomous vehicles holds immense potential to revolutionize traditional perceptions of automobiles, ushering in a new era of mobility wherein commuters can utilize travel time more efficiently, engaging in work-related tasks or other activities while

en route from point A to point B [67]. Moreover, autonomous vehicles have the capacity to enhance safety, decrease emissions, and optimize traffic flow, thereby potentially increasing road capacity [68].

Despite the technical feasibility of integrating autonomous vehicles into the transportation system, various non-technical factors, such as consumer confidence and legal considerations, along with the challenges of navigating vehicles in unpredictable and complex traffic conditions, present significant hurdles for the transportation industry [69]. This trend towards autonomous driving systems is not limited to the automotive sector but extends to other modes of transportation as well, including the maritime, aeronautical, and railway industries.

Alongside the development of autonomous driving systems, drones have emerged as another prominent trend in vehicle technologies. Initially utilized primarily in the military domain, drones are now increasingly being tested and deployed for a wide array of private and commercial applications [66,70].

### 2.5.2. Engine Technologies

In the realm of engine technologies, a significant trend shaping the industry is the shift towards the electrification of conventional drivetrains. Modern electric cars, now within reach of the average consumer, boast impressive ranges of 300 km or more on a single battery charge, with continual improvements in this area [71]. Despite advancements, concerns persist regarding the overall environmental impact of electric vehicles. Nonetheless, the market share of electric vehicles is steadily increasing and is projected to comprise 25 to 40% of new vehicle registrations globally by 2030 [72].

Beyond the automotive sector, the trend towards electrification is evident in other modes of transportation as well. In the maritime industry, electric ferries have been deployed on local routes, with recent orders indicating a growing interest in this technology [73]. Similarly, the aviation industry is exploring the potential of hybrid–electric propulsion technologies. Contrary to common belief, the electrification of drivetrains remains pertinent in sustainable rail transportation, with ongoing research focusing on clean fuel technologies. However, the main challenges lie in the costs associated with establishing additional infrastructure and converting existing drive technologies [74].

Another trend poised to significantly influence the development of sustainable engine technologies is hydrogen fuel cell technology. While hydrogen propulsion boasts vehicle-side efficiencies of around 45%, the energy-intensive production of hydrogen remains a concern. The International Energy Agency (IEA) predicts a market share of about 17% for fuel cell vehicles (FCVs) by 2050, with annual unit sales reaching 35 million [75].

### 2.5.3. Material Technologies

In the domain of material technologies, additive manufacturing stands out as a prominent trend that is rapidly gaining traction. Initially utilized primarily for prototyping purposes, 3D printing is now increasingly recognized for its significance in industrial mass production of individual components, particularly within the context of the emerging Industry 4.0 [76].

As highlighted by Wood [77], 3D printing technology has the potential to significantly disrupt traditional manufacturing practices by offering modular production alternatives, on-site manufacturing capabilities, and the flexibility to produce parts, tools, and even entire vehicles on-demand. Beyond the automotive sector, 3D printing is garnering increasing attention across various modes of transportation.

Another technology trend that is growing in popularity alongside 3D printing is lightweight construction. In the transportation industry, lightweight construction aims to conserve raw materials and energy by reducing vehicle weights [78]. Numerous concepts for lightweight vehicles have emerged in recent years, aligning with the objectives of energy optimization.

A similar objective to that of lightweight construction, particularly in terms of energy efficiency, is pursued through aerodynamic optimization. The primary goal of aerodynamic optimization is to design vehicles and infrastructures (such as tunnels) in a manner that minimizes air resistance, thus reducing the energy required for propulsion.

### 2.5.4. Infrastructure and Operating Technologies

With the progression of digitization and the growing Smart City initiative, intelligent transport systems (ITSs) are rapidly gaining prominence. ITS technologies are designed to enhance traffic flow and optimize infrastructure utilization by intelligently managing and coordinating various elements of traffic. Kantowitz and Le-Blanc [79] categorize ITS technologies into three types of communication: vehicle-to-infrastructure (V2I), infrastructure-to-vehicle (I2V), and vehicle-to-vehicle (V2V). Practical applications of V2I and I2V technologies include car park management, traffic control, usage-based billing, and navigation systems.

V2V communication technologies, such as adaptive cruise control, lane departure assistance, and active brake assistance, have become standard features in newer vehicle models, largely driven by advancements in the automotive industry towards (partially) autonomous driving. V2V communication establishes a wireless network where vehicles exchange messages containing information about their current status, including speed, location, and direction of travel. These data can then be processed using sophisticated algorithms to alert drivers about potential hazards ahead [80].

Another noteworthy technology in the realm of V2V communication is the development of swarm or collective intelligence. This concept suggests that individual actions can trigger intelligent behaviors within a community through communication and networking activities [81].

### 2.5.5. New Mobility Products and Services

As digitization progresses, accompanied by the technological advancements discussed earlier, a plethora of new mobility products and services have surfaced in recent years. According to Rauch [67], digital connectivity not only expands the range of mobility options but also introduces an entirely new dimension to mobility structures. The exchange of data among road users, vehicles, and infrastructure elevates mobility to a self-regulating system of real-time traffic management, on-demand availability, and seamless transitions between different modes of transportation.

One prominent trend in the realm of new mobility products and services is the emergence of sharing systems, driven by a fundamental shift in the concept of ownership. This shift entails a transition from owning transportation modes to utilizing them as needed [69].

Additionally, there has been a notable rise in the availability of Mobility as a Service (MaaS) products. MaaS integrates the entire transportation chain for users, encompassing trip planning, booking, and payment within a predominantly smartphone-based platform that accommodates all modes of transportation.

Furthermore, on-demand services have gained traction as a significant trend in the new mobility landscape. These services, ordered individually and as needed by consumers, have particularly proliferated in road-based transportation. Efforts to achieve fully autonomous operation of these systems are intensifying as research progresses [81,82].

## 3. Materials and Methods

The section presents the methodological assumptions of the authors' own research, including the research goal, research problems, research methods used, and the organization and course of the research.

### 3.1. Conceptual Assumptions

The aim of the study is to identify and analyze the challenges of the green transformation of transport in Poland. The identified challenges were determined based on the

following categories of factors: those internal to the country (strengths and weaknesses), as well as external factors relating to the broadly understood environment (opportunities and threats). Various methods were used in the research, depending on the phase (analysis of the subject literature, research implementation, preparation of results). In turn, the study used a number of methods depending on the stage of the study (preparation, implementation, processing of results), as shown in Figure 4.

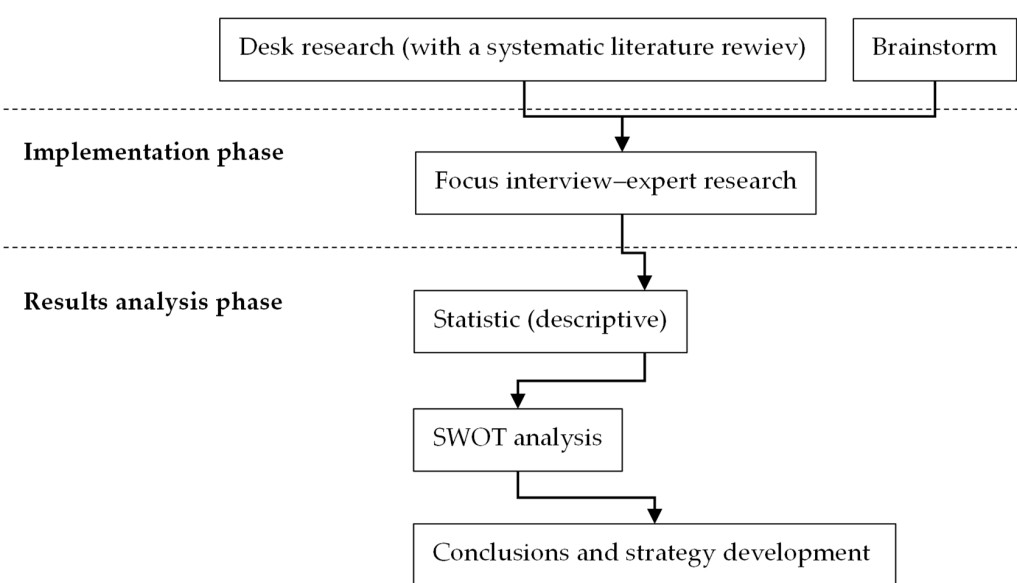

**Figure 4.** The scheme of the authors' research. Source: own study.

Below is a description and an indication of their use in individual stages of the study.

*Research preparation phase*

The authors formulated the following research problems:

- Q1: What are Poland's strengths in the context of the green transformation of transport (based on expert opinions)?
- Q2: What are Poland's weaknesses in the context of the green transformation of transport (based on expert opinions)?
- Q3: What opportunities can the green transformation of transport bring for Poland (based on expert opinions)?
- Q4: What threats does the green transformation of transport generate for Poland (based on expert opinions)?

**Desk research** entails analyzing accessible data sources, focusing on compiling, cross-verifying, and processing them. This analytical process forms the foundation for drawing conclusions regarding the researched issue [83,84].

**Brainstorming** is an activating work method that is based on generating solutions to given problems spontaneously. It involves working in a group, the aim of which is to generate ideas for finding the causes of the problem and solutions and selecting the best options [85].

Both methods were used to develop an expert interview form containing questions as well as suggestions for factors to be assessed (strengths, weaknesses, opportunities, and threats for the process of the green transformation of transport in Poland).

*Research implementation phase*

**Focus interview–expert research**, also known as the expert questionnaire, involves gathering research material through a structured questionnaire and obtaining responses

from participants selected by the researcher based on specific criteria. It is important to highlight that the expert interview represents a distinctive method that leverages the expertise and creativity of individuals who are authorities in a particular field [86]. The questions posed to respondents during the focus interview not only address facts or their attitudes but also aim to elicit explanations and predictions from them. Moreover, it is assumed that respondents who are professionally accomplished and possess expert knowledge can provide insightful analytical suggestions. Their professional expertise and ability to envision realistic scenarios enable them to offer valuable forecasts regarding the development of situations within specific domains of economic and social reality [87,88].

Expert interviews were conducted in January–February 2024.

*Results analysis phase*

**Focus interview–expert research**

The analysis of the test results was carried out in two parallel paths. The first one involves the use of elements of descriptive statistics, and the second one refers to SWOT analysis and determining the strategic position of the green transformation of transport in Poland.

**Statistical analysis included descriptive measures** such as arithmetic average, dominant value, minimum value, maximum value, and range [89]:

- The arithmetic average is the sum of the value of the variable of all units in the studied population divided by the number of units in the population.
- The dominant value is the value of the variable that is the most frequent (dominant, typical) in the studied community. The dominant is called a modal value or mode.
- The minimum value of recorded responses is in accordance with the adopted scale.
- The maximum value of the recorded responses is in accordance with the adopted scale.
- The range is the difference between the maximum and minimum.

**The SWOT** analysis is based on the classification of zero and green urban logistic factors into four categories:

- Advantages (internal positive).
- Disadvantages (inner negative).
- Opportunities (external positive).
- Threats (external negative).

An initial list of factors was selected in an earlier phase of the study, using desk research and brainstorming methods, and then supplemented during expert interviews. After identifying and assessing individual factors, a strategic position will be determined, i.e., a strategic framework will be developed for implementing the green transformation of transport.

Depending on the assessment of the strategic position, it is possible to find oneself in one of four areas that involve a specific approach (Figure 5).

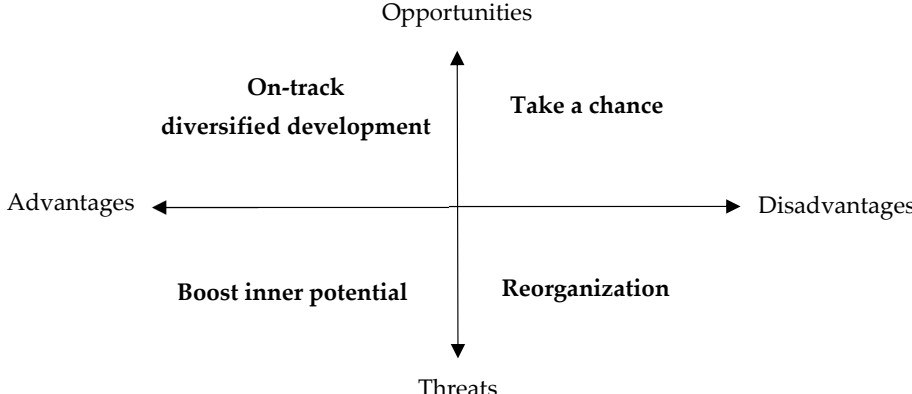

**Figure 5.** Strategic framework for green transformation of transport in Poland. Source: own study.

Assumptions of individual strategic approaches:

- **On-track** diversified development: this is a situation in which both advantages and opportunities prevail. This is a beneficial arrangement in which it is possible to implement various activities in the implementation of sustainable transport assumptions. It is recommended to maintain the current approach, strategic assumptions, and the implementation of policies related to eco-friendly and sustainable transport.
- **Take a chance**: this is a situation in which the disadvantages and opportunities of the environment prevail at the same time. It is recommended to take advantage of opportunities (especially those with the highest rating) while mitigating or correcting shortcomings. It is necessary to review and intensify actions that will ensure the development and sustainable development of eco-friendly transport.
- **Boost inner potential**: this is a situation in which the benefits and risks are weighed. The recommendation concerns counteracting threats by using numerous and highly rated assets. Opportunities to further strengthen strengths and lead the development of clean transport should be considered.
- **Reorganization**: this is the least favorable situation, as both disadvantages and threats prevail. It is recommended to revamp the approach to the development of eco-friendly and sustainable transport. Possible actions include updating/developing national and regional development strategies, developing policies, financing appropriate activities, and developing actions aimed at minimizing identified defects.

It is worth pointing out that the authors used a fundamentally classic approach to SWOT analysis; however, in the context of the strategic framework (the assessment of strategic position), due to the specificity of the analyses (in reference to the multi-dimensional process of socio-economic changes), the authors proposed an original approach to identifying strategic opportunities that corresponds to the specificity of such sectoral analyses.

### 3.2. Selection of Experts for the Research

The technique of purposeful sampling was used. The selection criteria for the study were extensive practical experience in the field of eco-friendly, low-emission, sustainable transport solutions and technologies (for various types of transport: road, rail, air, inland river, and sea), manifested by participation in the development of strategic assumptions for the implementation of this type of solution in enterprises or in a given region, in international projects related to the implementation of sustainable transport technologies, in the development of policies, or in a research agenda in this area at the European Union level. The request to participate in the study was sent to experts from various regions of Poland: scientists, industry experts, employees of research institutes, and experts from the real economy sector (from enterprises related to transport activities and the automotive industry). A total of 51 experts took part in the study. Due to the subject of the study, the number of participants was not selected as representative. Experts have deep knowledge in their fields, which means that they provide detailed, precise, and reliable information on the topic under discussion. Thanks to their knowledge and experience, experts present unique perspectives and points of view. Therefore, the analysis results constitute a unique added value to the current state of knowledge.

### 3.3. Tool Description

The interview questionnaire consisted of several parts:

- Details: questions about the size of the company where the expert is employed, position held, experience;
- Assessment of Poland's individual strengths in the context of the green transformation of transport;
- Assessment of Poland's individual weaknesses in the context of the green transformation of transport;
- Assessment of individual opportunities for Poland in relation to the green transformation of transport;

- Assessment of specific threats to Poland in relation to the green transport transformation.

## 4. Results

A total of 51 deliberately selected experts took part in the study. The criterion for selecting experts was extensive practical experience in the development and implementation of solutions and technologies for eco-friendly, low-emission, sustainable transport. Almost half of the experts (47.06%) were employed in large enterprises (employing more than 249 employees). Moreover, 19.61% were employed in medium-sized commercial organizations (employing 50 to 249 employees), 25.49 were employed in commercial micro-organizations (employing less than 10 employees), and the remaining 7.84% were employed in small entities (employing 10 to 49 employees), as shown in Figure 6.

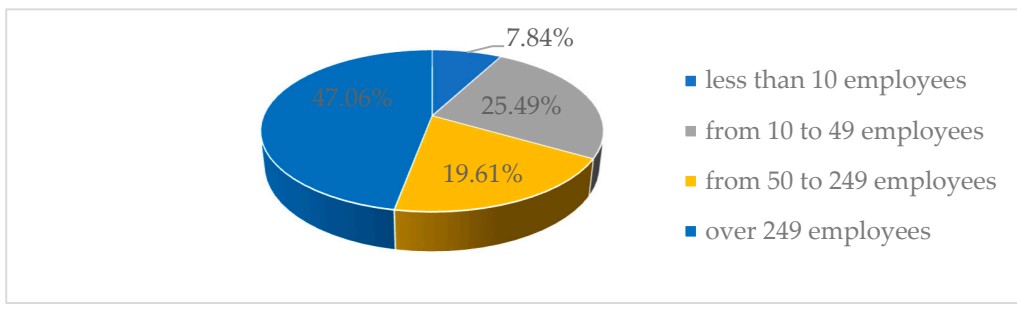

**Figure 6.** The size of the company employing the expert. Source: own study.

Most of the experts (50.98%) were employed as specialists. Additionally, 15.69% were employed as managers, and supervisors and directors were 11.76% each. Moreover, business owners constituted 7.84%, and the remaining 1.96% were management (Figure 7).

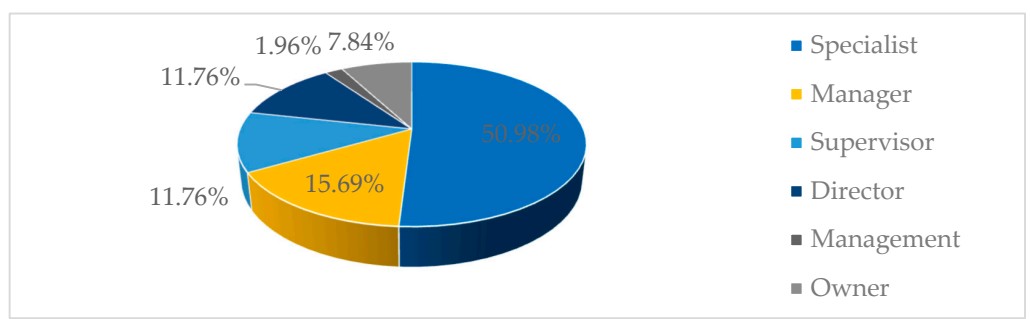

**Figure 7.** Expert workplace. Source: own study.

Referring to the professional experience of experts in the current position, the largest percentage (45.10%) has been employed for more than 5 years, 35.29% were employed for 1 to 3 years, and 19.61% were employed for 3 to 5 years (Figure 8).

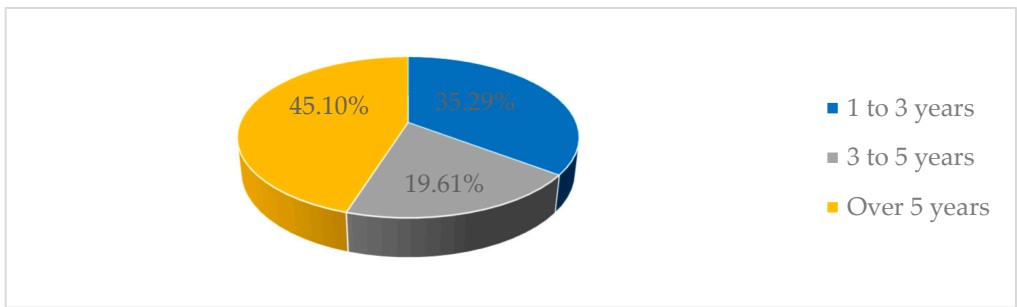

**Figure 8.** Professional experience in the last position in years. Source: own study.

Experts also assessed their own knowledge and experience on a scale of 1 to 5 (where 1 means a low level of knowledge and experience, and 5 means a high level of both knowledge and experience). The largest percentage indicated a grade of 4 (33.33%), 29.41% indicated a grade of 5, 23.53% indicated a grade of 3, and the remaining 13.73% indicated a grade of 2. There was no answer regarding the level 1 assessment. The average grade was 3.67, and the median was 4 (Figure 9).

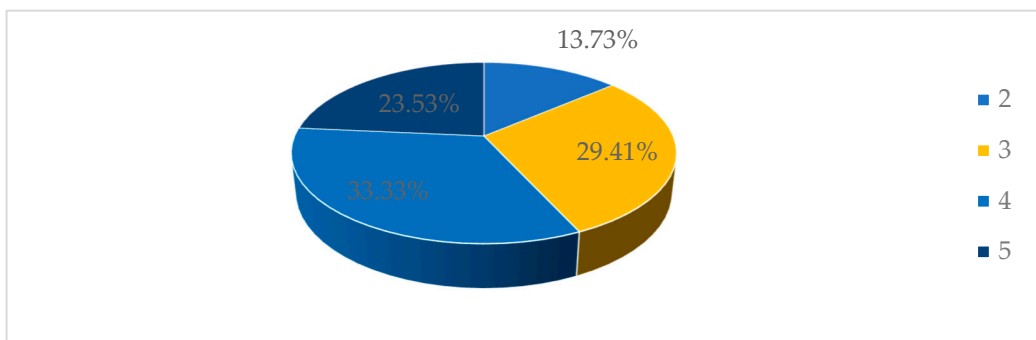

**Figure 9.** Self-assessment of expert knowledge and experience. Source: own study.

Analyzing the individual components relating to the SWOT method, in the context of strengths, the most important were a young society that adapts faster to new technological solutions (average 4.00). This is due to the fact that Poland has a relatively young society that is open to new technologies and changes in the way we use transport. Young people are often more willing to accept innovative solutions such as electric vehicles and car sharing services, which can accelerate the adoption of green transport alternatives, as well as the potential to use mobile applications and online platforms to promote sustainable forms of transport, such as public transport, city bikes, and carsharing.

The next strength is in financing and the availability of EU funds and aid funds that can support investments in environmentally friendly transport infrastructure (average 3.94). In this approach, it is possible to use available European Union funds and other sources of financing that are intended for projects related to the modernization of transport infrastructure towards green solutions. EU funding can significantly reduce the costs of investment in electromobility infrastructure, such as chargers for electric vehicles or the expansion of the public transport network.

The third most important strength is the technological potential and support of innovation in electric transport and alternative energy sources (average 3.55). Poland has advanced technologies and experts in the fields of electric transport and alternative energy sources. There are many research centers, institutes, and enterprises involved in the development of technologies related to green mobility and active support for innovative startups and enterprises operating in the electric transport sector, which promotes the development of innovative solutions and products on the market. This has the potential to develop charging infrastructure for electric vehicles and public transport networks based on low-emission vehicles, such as electric buses and trams (Table 1).

In terms of weaknesses, we can primarily distinguish existing transport infrastructure requiring significant investment to support the development of electric vehicles and charging infrastructure (average 4.24). It should be noted that the existing road and urban infrastructure requires modernization and adaptation to the needs of electric vehicles, which may be expensive and time-consuming. The need to expand the charging network, including installing chargers on streets, public and private parking lots, and gas stations, may require significant financial outlays and logistical resources.

The next weakness is the concerns relating to the purchase and use of electric vehicles of citizens (average 4.10). The high costs of purchasing electric vehicles may be a significant barrier for consumers, especially in the context of the lower income of many Polish residents.

A lack of experience or uncertainty about the performance, efficiency, and long-term viability of electric vehicles may inhibit consumers' willingness to purchase them.

**Table 1.** Analysis of the strengths of the green transformation of transport in Poland. Source: own study.

| Factor | Average | Dominant Value | Maximal Value | Minimal Value | Range |
|---|---|---|---|---|---|
| Financing: availability of EU funds and aid funds that can support investments in environmentally friendly transport infrastructure | 3.94 | 4 | 5 | 1 | 4 |
| Experience in the implementation and operation of electric public transport | 3.25 | 3 | 5 | 1 | 4 |
| Experience in the implementation and operation of hydrogen-powered public transport | 3.06 | 3 | 5 | 1 | 4 |
| Taking initiatives in the field of greening last mile logistics (e.g., cargo bikes, implementation of eco-friendly means of transport by KEP industries, carriers) | 3.37 | 3 | 5 | 1 | 4 |
| Implementation of the concept of smart cities, including solutions for optimizing the flow of goods and people in the city | 3.27 | 3 | 5 | 1 | 4 |
| Introduction of clean transport zones in cities | 3.10 | 3 | 5 | 1 | 4 |
| Technological potential and supporting innovation in electric transport and alternative energy sources | 3.55 | 4 | 5 | 1 | 4 |
| Availability of electric vehicles as part of carsharing services | 3.22 | 3 | 5 | 1 | 4 |
| A young society that adapts faster to new technological solutions | 4.00 | 4 | 4 | 4 | 0 |

The third most important weakness is the limitations and difficulties in the use and operation of electric vehicles (average 4.06). The shortage of charging infrastructure in some regions of Poland may make the everyday use of electric vehicles difficult, especially during long-distance trips. The limited range of electric vehicles and charging times can be an inconvenience for users, especially if charging is not possible at home or at work. The lack of sufficient electric vehicle models available on the market may limit consumer choice and lead to competition in the market, which will impact prices (Table 2).

Among the opportunities, the most important was clean air in cities (average 5.0). Growing public awareness of air pollution problems may put pressure on policy makers and institutions to take action towards the green transformation of transport. Initiatives limiting the use of combustion vehicles in city centers and promoting public transport and electric and eco-friendly forms of transport can contribute to improving air quality.

The second most important opportunity was the growing ecological awareness of society to promote acceptance (average 4.08). More and more people are aware of the negative impact of road transport on the environment and human health, which may lead to greater support for green mobility initiatives. Active educational, informational, and promotional campaigns can contribute to raising public awareness of the benefits of eco-friendly forms of transport.

The third most important opportunity was the growing demand for public transport and for more eco-friendly forms of transport, such as electric bicycles and carsharing (average 3.96). The growing number of people using public transport can support the development of public transport infrastructure and encourage investment in more eco-friendly vehicles, such as electric buses or trams. The increasing demand for eco-friendly forms of transport, such as electric bicycles and carsharing, may stimulate the development of these sectors, which will contribute to reducing air pollution and road congestion (Table 3).

**Table 2.** Analysis of the weaknesses of the green transformation of transport in Poland. Source: own study.

| Factor | Average | Dominant Value | Maximal Value | Minimal Value | Range |
|---|---|---|---|---|---|
| Existing transport infrastructure requiring significant investment to support the development of electric vehicles and charging infrastructure | 4.24 | 5 | 5 | 1 | 4 |
| The dominance of fossil fuels in the structure of energy production makes it difficult to move to more eco-friendly energy sources | 3.92 | 4 | 5 | 2 | 3 |
| Underdeveloped electric vehicle market (little interest in electric vehicles) | 3.92 | 4 | 5 | 1 | 4 |
| Limitations and difficulties in the use and operation of electric vehicles | 4.06 | 4 | 5 | 2 | 3 |
| Concerns relating to the purchase and use of electric vehicles of citizens | 4.10 | 4 | 5 | 2 | 3 |
| Resistance from traditional industries, which may delay changes towards the green transformation of transport | 3.94 | 4 | 5 | 2 | 3 |
| Lack of political stability | 3.78 | 4 | 5 | 1 | 4 |
| Low level of innovation | 3.40 | 3 | 5 | 1 | 4 |
| No strategic assumptions or planning of the green transformation process | 3.88 | 4 | 5 | 1 | 4 |
| Relatively slow green transformation of house and apartment heating | 4.00 | 4 | 4 | 4 | 0 |
| Wages have not become significantly higher compared to rampant inflation, which means that society will feel the impact of the green transformation more | 4.00 | 4 | 4 | 4 | 0 |

**Table 3.** Analysis of the opportunities of the green transformation of transport in Poland. Source: own study.

| Factor | Average | Dominant Value | Maximal Value | Minimal Value | Range |
|---|---|---|---|---|---|
| Renewable energy potential that can be used in the development of transport electrification | 3.78 | 4 | 5 | 1 | 4 |
| Growing ecological awareness of society may promote acceptance and adaptation of green transport technologies | 4.08 | 4 | 5 | 2 | 3 |
| Functioning within the European Union and adapting legal conditions to the implementation of the assumptions of the green transformation | 3.75 | 4 | 5 | 1 | 4 |
| Possibility to create public–private partnerships to develop charging infrastructure and other eco-friendly transport solutions | 3.59 | 4 | 5 | 1 | 4 |
| There is an opportunity to develop innovative business models that promote sustainable transport | 3.82 | 4 | 5 | 1 | 4 |
| Growing demand for public transport and for more eco-friendly forms of transport, such as electric bicycles and carsharing | 3.96 | 4 | 5 | 2 | 3 |
| Stimulating economic growth and development by creating new branches of the economy | 3.92 | 4 | 5 | 2 | 3 |
| Participation of Polish organizations in international research and development projects aimed at developing low-emission transport technologies and solutions (e.g., under the Horizon Europe program) | 3.80 | 4 | 5 | 1 | 4 |
| Increasing the country's innovativeness | 3.90 | 4 | 5 | 1 | 4 |
| Clean air in cities | 5.00 | 5 | 5 | 5 | 0 |

The last aspect analyzed was the threats to the green transformation of transport in Poland. The most important were problems with the disposal of batteries in vehicles and the short life of vehicles because, if destroyed, the vehicle must be disposed of (average 5.0.). The increase in the number of electric vehicles leads to an increase in waste batteries, which creates challenges in the disposal and recycling of batteries as they are a source of toxic waste. The short operating life of electric vehicles due to the limited durability of the batteries may lead to an increased amount of waste related to the disposal of entire vehicles.

Another threat was the methods of generating electricity and the constant increase in electricity fees (average 5.00). Poland relies largely on coal for its energy sources, which may contribute to increased greenhouse gas emissions unless measures are introduced to increase the share of renewable energy sources. An increase in electricity fees may affect the costs of using electric vehicles and be an inhibitory factor for potential users.

The third most important threat was the social resistance to price increases and additional fees for the use of combustion-powered vehicles (4.35). The introduction of additional fees, taxes, or restrictions for combustion vehicles may meet with public resistance, especially in the case of lower-income social groups, who may be more sensitive to price changes. There may be resistance from the industrial sector, especially those related to the production of combustion vehicles, which may oppose the introduction of restrictions and regulations favoring electric vehicles (Table 4).

**Table 4.** Analysis of the threats of the green transformation of transport in Poland. Source: own study.

| Factor | Average | Dominant Value | Maximal Value | Minimal Value | Range |
|---|---|---|---|---|---|
| Social resistance to changes in transport and lifestyle | 3.82 | 4 | 5 | 1 | 4 |
| Social resistance to price increases and additional fees for the use of combustion-powered vehicles | 4.35 | 5 | 5 | 1 | 4 |
| Social resistance to bearing the costs of the green transformation of transport | 4.34 | 5 | 5 | 1 | 4 |
| Reducing emissions of greenhouse gases and pollutants by Poland and Europe does not affect global emissions and will not contribute to stopping climate change | 3.65 | 4 | 5 | 1 | 4 |
| Loss of competitiveness of the Polish economy by imposing fees and taxes on activities related to the emission of carbon dioxide and other pollutants | 3.86 | 4 | 5 | 1 | 4 |
| Increased costs of producing energy from fossil fuels (through additional fees and taxes) | 3.96 | 4 | 5 | 1 | 4 |
| Lack of technologies and solutions ensuring continuous and stable energy production from renewable energy sources | 3.90 | 4 | 5 | 1 | 4 |
| Impoverishment of society caused by the need to pay fees related to the green transformation | 3.88 | 4 | 5 | 1 | 4 |
| Lack of developed charging infrastructure for electric vehicles in the country | 4.14 | 4 | 5 | 1 | 4 |
| Political instability that may lead to changes in government priorities | 3.94 | 4 | 5 | 1 | 4 |
| Lack of a uniform approach to green transport issues in the European Union, which may lead to difficulties in harmonizing regulations and standards | 3.90 | 4 | 5 | 1 | 4 |
| Lack of solutions and technologies for the disposal and recycling of electric vehicle parts and vehicles | 4.12 | 5 | 5 | 1 | 4 |
| Problems with the disposal of batteries in vehicles and the short life of vehicles because, if destroyed, the vehicle must be disposed of | 5.00 | 5 | 5 | 5 | 0 |
| High fire hazard if an electric vehicle catches fire | 4.00 | 4 | 4 | 4 | 0 |
| Methods of generating electricity and the constant increase in electricity fees | 5.00 | 5 | 5 | 5 | 0 |
| Too strong a push for the development of zero-emission vehicles in favor of hybrid vehicles | 4.00 | 4 | 4 | 4 | 0 |
| Lack of belief in the appropriateness of using electric drives if the electricity does not come from renewable energy sources | 3.00 | 3 | 3 | 3 | 0 |

Based on the analyses carried out in the context of the strengths, weaknesses, opportunities, and threats related to the green transformation, transport in Poland, and the opinions of experts, an assessment of the strategic position for this process was made. To assess the strategic position, it is necessary to parameterize individual areas in the SWOT method. The arithmetic mean was used for this purpose (in each of the four aspects of the method). At the same time, positive ratings were assigned to "positive" factors, i.e., strengths and opportunities, and negative values were assigned to "negative" factors, i.e., weaknesses and threats (Table 5).

**Table 5.** Average ratings in individual areas of the SWOT analysis. Source: own study.

| Strengths | Weaknesses |
|---|---|
| 3.42 | −3.93 |
| Result of internal assessment of factors: −0.51 | |
| **Opportunities** | **Threats** |
| 3.96 | −4.02 |
| Result of assessment of environmental factors: −0.06 | |

A visualization of the strategic position of the green transformation of transport in Poland is presented in Figure 10.

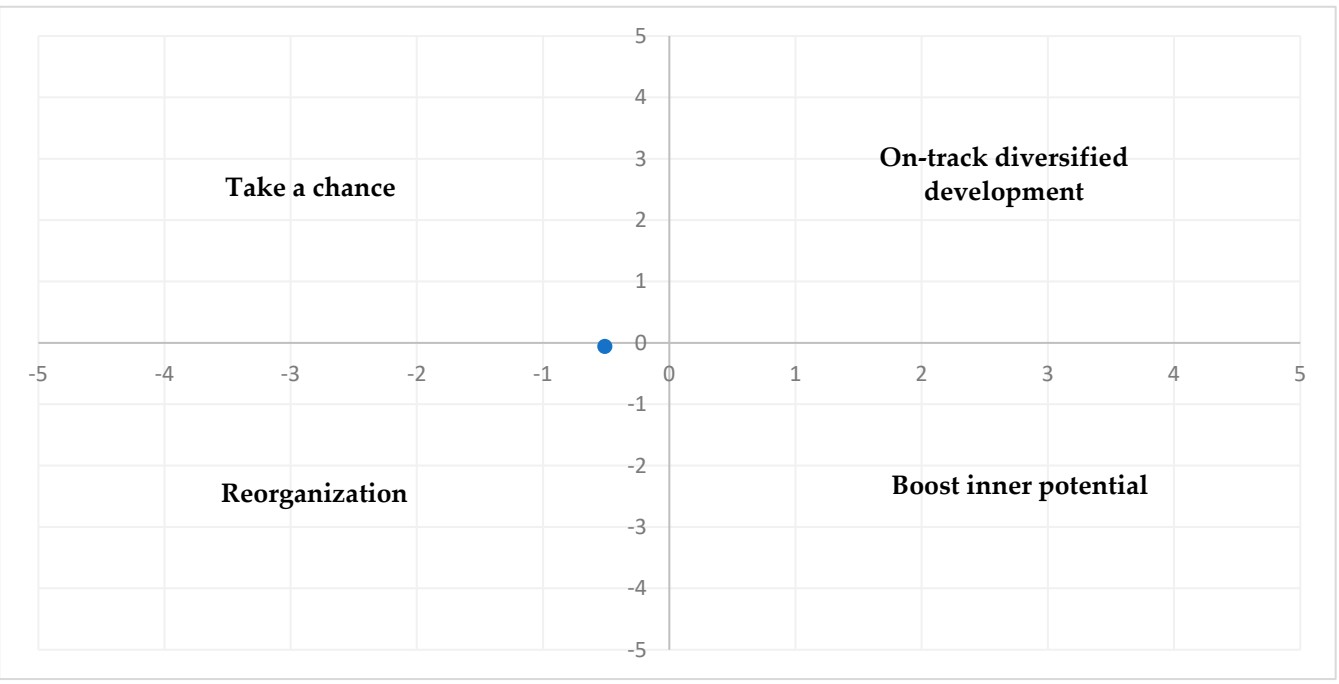

**Figure 10.** Assessment of the strategic position of the green transformation of transport in Poland. Source: own study.

Based on the analyses performed, it can be indicated that the process of the green transformation of transport in Poland, in the context of the strategic framework, was placed in the least favorable approach: reorganization. This assumes the need for a thorough reconstruction of the approach to the development of eco-friendly and sustainable transport. It is a situation in which disadvantages and threats dominate, requiring coordinated action at various levels. It will be necessary to conduct a detailed analysis of existing national and regional development strategies, identifying their weak points and areas requiring improvement. At the same time, broader financing of pro-ecological activities, support

for research and innovation, and an educational program for society should be taken into account.

## 5. Discussion

It should be noted that analyses relating to the possibility and effectiveness of greening transport in Poland are being conducted. At the same time, similar conclusions or research results can be pointed out. This process will be extremely demanding, and mitigating potential threats will require a complex approach both at the level of administrative authorities and changes in social habits.

The Polish Organization of Petroleum Industry and Trade has developed three possible scenarios for this process [2]:

- Passive scenario: slow, formal changes, mainly forced by Poland's obligations towards the European Union. Continuous search for ways to "bypass" external requirements to popularize green transport. The prevailing threats include loss of competitiveness of the Polish economy, problems of the Polish transport sector on the markets of Western Europe, and difficult access of Polish cars to roads in other countries (e.g., possible entry bans for high-emission vehicles).
- Accelerated change scenario: quick and radical changes leading to a significant increase in the share of green fuels in Poland's transport mix. They ignore the important interests of many stakeholder groups, including owners of cars with combustion engines. In this approach, the following threats were identified: the risk of social dissatisfaction and tension, especially in groups that lose out due to rapid changes, and the increasing scale of poverty in groups excluded from communication. Opportunities included, for example, the rapid development of new branches of the economy related to green energy and the possibility of obtaining funds from EU funds and financial institutions for a profound transformation of transport.
- Sustainable development scenario: the changes combine both Poland's obligations related to membership in the European Union, regarding the share of green energy in the transport energy mix, and the interests of social groups that may lose out from too quick and radical changes. In this approach, the prevailing opportunities are the development of new industries related to ecology and green transport, the increasing importance of Poland as a supplier of technological solutions for green transport, and the effective use of investment outlays for the green transformation. Examples of threats are attempts by hydrocarbon suppliers to slow down the transformation by lowering prices and different perceptions of the "sustainable approach" in Poland in Western Europe (Polish changes may be perceived as too conservative).

In the opinion of Bęben [90], transport transformation will be the greatest challenge that Poland will face in the coming years. Due to the high costs, it is extremely important that the changes do not affect the poorest part of society or lead to communication exclusion. In Poland, we must look for and focus on solutions that take into account the specificity of the country and make maximum use of resources. An example is increasing the share of modern types of biofuels in the fuel mix, which will also benefit Polish agriculture. "Greening transport" will also not be possible without rebuilding, or rather expanding, the energy system towards increasing the share of energy from renewable sources. Instead of energy from fossil fuels, currently coming mainly from imports, low- and zero-emission vehicles will be powered by renewable energy produced in Poland.

Moreover, it can be pointed out that the creation of a low-emission transport system requires the transformation of the entire socio-technical system. In themselves, behavioral changes of individual consumers or the availability of technology may not be enough to change the system, or these changes will last for a longer time, which may lead to losses for entities in this sector. From the perspective of users and demand for zero-emission vehicles, there is a need to reformulate business models, change user habits, develop appropriate informational campaigns, increase the availability of charging infrastructure, and take into account the increasing importance of sustainable development aspects and the growing

consumer pressure to strive for climate neutrality in the entire value chain. System transformation requires overcoming resistance to innovation and barriers to change. The change requires overcoming a number of barriers: technical and economic (dependencies resulting from, for example, investments in equipment, skills, and knowledge); institutional and political (connections, norms, standards); social and cognitive (habits resulting from functioning in the existing system, lack of openness to innovation, habits to a specific lifestyle, and consumption patterns). All this makes radical innovation particularly difficult [91,92].

Moreover, it is possible to indicate several examples of the implementation of "green transport" solutions in various places around the world.

In 2020, the share of green fuels in the fuel mix used in transport in Sweden was 31.85%, which gave this Scandinavian country first place in the European Union (the share of green energy in transport was three times higher than the EU average). According to the Swedish Climate Policy Council, $CO_2$ emissions from transport decreased by approximately 20% in 2017 compared to 2010. Three factors underpin Sweden's success in greening transport: social changes promoting efficient transport, electrification, and increasing the share of biofuels. Tools were used at both national and local levels. One of the controversial but effective solutions used in Sweden was to limit the possibility of creating parking lots in places with high population density where there is no space for parking spaces. As a result, residents of some housing estates, deprived of the opportunity to park their cars nearby, give up their own cars completely. An alternative in such situations is, among others, car-sharing [93].

In 2008, the state of California in the USA passed Senate Bill 375. Its goal was to significantly reduce VKT (the number of kilometers traveled). California's activities consist of administrative, economic, and social-informational tools. To reduce car traffic, among other measures, road reconfiguration programs were implemented (reducing their size). The expansion of transit routes and investments in improving the quality of public transport also began. Emphasis was also placed on the construction of housing estates equipped with full "social infrastructure" so as not to force residents, for example, to drive their children longer distances to school or to go shopping and to their workplace. California authorities have also initiated the construction of bicycle paths and pedestrian sidewalks. However, the basic economic tool is the fuel tax, which was introduced in California in 2000. In addition, an additional excise tax was introduced on the sale of cars with diesel engines. At the local level, Californians are also introducing incentives for remote work [2].

Another implementation example is from the city of Odense (in Denmark), the pioneer city in Northern Europe, which has introduced a cutting-edge rain sensor at a traffic light along their Super Bike Highway. This innovation allows the intersection to detect rainfall, resulting in cyclists enjoying extended green light cycles by up to 20 s. This is essential to allowing cyclists to spend less time waiting at red lights, especially during inclement weather. The rain sensor operates in tandem with two motion detectors installed on the traffic light. These detectors enable the system to identify approaching cyclists within a 70 m radius of the intersection, automatically prolonging the green light duration for their convenience. A mounted box on the traffic light serves to inform cyclists about the system's operation; it illuminates when the system activates. While cars traveling along the Super Bike Highway may experience a slightly prolonged red light, it is a minor inconvenience compared to the benefits for cyclists. After all, they are the ones who would otherwise have to contend with being drenched in rain. The overarching goal of this system is to encourage cycling as a preferred mode of transportation, with the rain sensor representing a clever technological enhancement aimed at improving the daily commute for cyclists [94].

Another example, also from Denmark, from the city of Copenhagen, is the Bus Rapid Transit (BRT) solution, which has proven to be an effective and economical mode of public transportation, gaining popularity in numerous cities worldwide. Copenhagen has dedicated substantial effort over the years to enhance bus transit efficiency for the benefit of its millions of passengers. Various initiatives have been implemented in the city to facilitate smoother bus travel, including the establishment of dedicated bus lanes that

receive priority access to green lights ahead of other traffic. In Valby, these traffic signals have been equipped to detect buses on the streets and facilitate their passage through green lights. This is achieved through the utilization of 66 cameras capable of anonymously identifying all motor vehicles, along with a GPS system installed in the buses. Initial outcomes indicate that individual bus routes have reduced travel times by up to two minutes during rush hours over short distances. It is anticipated that motorists and cyclists in the area will experience a 20% decrease in travel time [95].

Zhang et al. [82] conducted analyses in Shenzhen (China) relating to the estimation of carbon dioxide emission reductions through the use of digital technologies in transport. According to their research findings, the residents of Shenzhen managed to decrease $CO_2$ emissions by approximately 1.2 million tons over a span of three years (2019–2021) by utilizing the subway, facilitated by the Tencent Map green transportation platform. Furthermore, the adoption of the Tencent Map Bus Code digital tool in the city led to a reduction of nearly 27,000 tons of $CO_2$ emissions by Shenzhen citizens within 17 months of its launch.

At the same time, it is worth emphasizing in a general sense that empirical evidence suggests that the systematic expansion of public transportation networks, including subways, light rail, and Bus Rapid Transit (BRT), serves as a viable strategy to conserve energy and mitigate carbon emissions in urban transportation systems across both developed and developing nations [96,97]. Comparatively, the energy consumption of cars is three times higher than that of buses when covering the same distance. Enhancing non-motorized transportation infrastructure through the establishment of public transit systems contributes to the development of an efficient, environmentally friendly urban transportation network [98].

Green transformation involves enhancing the efficiency of energy resource utilization, minimizing pollutant emissions, reducing environmental impact, optimizing work efficiency, and fostering sustainable development, thereby achieving a favorable balance between economic prosperity and environmental conservation [99]. According to Li and Lin [100], technological advancements serve as the cornerstone for enhancing energy efficiency and conservation, thereby driving green transformation. This perspective is echoed by Söderholm [101], who emphasizes the importance of promoting sustainable technology-driven development characterized by production and consumption models with significantly reduced adverse impacts on the natural environment, including mitigating climate change effects. Ngai and Pissarides [102] assert that ongoing technological innovation serves as a continuous driver of sustainable economic development, playing a pivotal role in optimizing resource utilization efficiency and modernizing industrial infrastructure.

Nevertheless, it is essential to reiterate, as indicated by Stern and Rydge [45], that technological advancement alone may not determine the pace of the transformation process. Equally significant is the context, system, and environment in which these technologies are evolving. In more advanced economies, there tends to be a greater societal willingness to invest in such solutions, whereas energy transformation in countries trailing behind leaders may progress more slowly for various reasons.

## 6. Conclusions

The challenges related to the green transformation of transport in Poland are related, in particular, to the need to adapt to the requirements of reducing carbon dioxide emissions and modernizing the energy sector. The transition to more eco-friendly forms of transport is becoming one of the main challenges for the Polish economy, especially in the context of increasing financial outlays and taking into account social interests.

The study involved 51 deliberately selected experts with extensive practical experience in the development and implementation of solutions and technologies for eco-friendly, low-emission, sustainable transport. In terms of SWOT analysis, the strengths included a young society that is adaptable to new technologies and the availability of EU funds for

environmentally friendly infrastructure. The weaknesses encompassed existing transport infrastructure, concerns about electric vehicle adoption, and limitations in their use. The opportunities highlighted clean air, a growing ecological awareness, and the demand for public transport. The threats included issues with battery disposal, electricity generation methods, and societal resistance to change. The strategic position assessment resulted in a net negative score for internal factors ($-0.51$) and environmental factors ($-0.06$), indicating challenges in the transformation process. The approach aligns with "reorganization", demanding a comprehensive reconstruction of the development approach. This requires coordinated actions, including detailed analysis of existing strategies, broader financing, research support, and public education programs. Ultimately, the study identifies the need for substantial efforts to effectively navigate the green transformation of transport in Poland.

The following strategic recommendations for the green transformation of transport in Poland have been developed:

- Update of National and Regional Development Strategies:
  - The need for comprehensive updates to national and regional development strategies, considering the priorities of sustainable transport.
  - Introduction of flexible development policy frameworks enabling adaptation to changing conditions and needs.

- Development of Pro-Ecological Policies:
  - Creation and implementation of policies conducive to sustainable transport development, promoting the use of alternative transportation means such as public transport, bicycles, and pedestrian travel.
  - Establishment of financial and regulatory incentives for the private sector to invest in eco-friendly transport solutions.

- Financing of Pro-Ecological Actions:
  - Increase in financial allocations for investments related to the development of environmentally friendly transport infrastructure.
  - Establishment of financial support mechanisms for businesses and institutions willing to modernize their fleets towards more eco-friendly solutions.

- Minimization of Identified Weaknesses:
  - Identification of major flaws in the current transport system and development of actions aimed at their minimization.
  - Focus on reducing greenhouse gas emissions, improving energy efficiency, and mitigating the negative impact of transport on the natural environment.

- Support for Research and Innovation:
  - Encouragement of scientific research and innovation in the field of eco-friendly transport technologies.
  - Creation of conditions fostering cooperation between the public, private, and academic sectors for the development and implementation of new solutions.

- Education and Public Awareness:
  - Promotion of eco-friendly education and raising public awareness about the benefits of sustainable transport.
  - Organization of informational and educational campaigns mobilizing society for active participation in the transport transformation process.

- International Cooperation:
  - Collaboration with other countries and international organizations to exchange experiences and best practices and adopt a joint approach to green transport development.
  - Active participation in international programs and initiatives aimed at reducing greenhouse gas emissions and promoting sustainable transport.

It should be noted that an effective green transformation of transport in Poland requires an integrated approach, including infrastructure investments, financial incentives, social education, and dialog and cooperation between various stakeholders. It is necessary to adopt a comprehensive strategy that will take into account all the above-mentioned challenges. This process requires a balanced approach that takes into account environmental, social, and economic goals and takes into account weaknesses and threats. The following actions should be taken:

- It is necessary to significantly increase investment in the expansion of charging infrastructure for electric vehicles, including the construction of charging stations in strategic locations such as public parking lots, workplaces, and shopping centers.
- A plan is required to phase out fossil fuels in favor of renewable energy sources such as solar, wind, and biomass. Government support and financial incentives for investment in renewable energy sources will be crucial.
- It is necessary to encourage the purchase of electric vehicles by introducing financial incentives, such as tax breaks, subsidies for the purchase of electric vehicles, and reducing electricity prices for electric vehicle users.
- Education and training for drivers and mechanics on the operation and maintenance of electric vehicles are required. Also, expanding the charging infrastructure and improving the availability of spare parts for electric vehicles will be crucial.
- Conducting educational and informational campaigns that will dispel fears and uncertainties regarding electric vehicles and promote the benefits of their use for the environment and owners' wallets.
- Dialog and cooperation are required between the government, traditional businesses, and the transport sector to identify ways to support and accelerate transformation while minimizing negative impacts on existing industries.
- Developing a long-term strategy for the green transformation of transport, taking into account political stability and promoting innovation through incentives for enterprises and the research and development sector.
- Developing and implementing a comprehensive strategy for the green transformation of transport, which will include goals, priorities, and specific actions, as well as the monitoring and evaluation of progress.
- Promoting investments in more eco-friendly heating sources, such as heat pumps, solar panels, and biomass boilers, through financial incentives, subsidy programs, and public education.
- Introduction of measures mitigating the effects of the green transformation for people with lower incomes, such as financial support programs for the purchase of eco-friendly vehicles or subsidies for the thermal modernization of buildings.
- Conducting educational campaigns that explain to society the benefits of the green transport transformation, such as improving air quality, reducing greenhouse gas emissions, and reducing noise.
- Introduction of financial incentive systems for people who decide to purchase and use low-emission vehicles, as well as subsidy programs for replacing old vehicles with more eco-friendly ones.
- Developing fair and equivalent financial solutions that minimize the burden on lower-income populations, e.g., through tax exemptions for lower-income people or subsidy programs for green transport solutions.
- International cooperation and Poland's involvement in global actions to reduce emissions are required through participation in international climate agreements and promoting actions at the European and international level.
- A balanced approach to emission taxation is needed that will take into account both environmental goals and the competitiveness of the economy, e.g., by implementing emission trading systems while supporting enterprises in adapting to new regulations.
- Phase out subsidies for fossil fuels and redirect these funds to support energy production from renewable sources such as solar, wind, and biomass.

- Intensive investments are required in the research and development of renewable energy technologies and support for enterprises producing and implementing these technologies.
- Introduction of financial support programs for people with lower incomes, such as tax breaks, subsidies for replacing old devices with more energy-efficient ones, and subsidy programs for eco-friendly transport solutions.
- Intensive investment is required in the development of charging infrastructure, including the construction of charging stations in strategic locations such as public parking lots, gas stations, and shopping malls.
- Striving to maintain political consensus on issues related to the green transformation of transport through intergovernmental dialog and the involvement of all political parties in the decision-making process.
- Poland's active participation in decision-making processes at the EU level to ensure the harmonization of regulations and standards regarding the green transformation of transport.
- Investment is required in the research and development of technologies related to the disposal and recycling of electric vehicles and the promotion of innovation in this area.

To sum up, the green transformation of transport in Poland must be based on a comprehensive approach, taking into account various areas of activity and cooperation between the public and private sectors and civil society. Limitations of this study result from the scope of the focus group interview, of which the 51 experts are not representative. At the same time, the selected research parameters, such as strengths, weaknesses, opportunities, and threats, were selected based on an analysis of the literature on the subject, brainstorming, and interviews with experts. Therefore, future research should include the widest possible sample of involved experts, not only from the real economy but also from the scientific community. Further research should also include other research methods, such as Delphi, or the analysis of strategic documents at the national and EU level, including research and development programs. At the same time, future studies should also cover individual regions of Poland and their development differences in the context of preparation for the "greening of transport".

Additionally, future research could focus on the following:

- Analyzing the potential impact and applications of new technologies (e.g., electric vehicles, renewable energy, etc.) to facilitate the practical implementation of the clean transport transformation;
- Analysis of the energy potential of various renewable sources in the context of meeting transport needs;
- Analysis of fast-charging technology and the impact on the energy network;
- Analysis of potential benefits and challenges related to the integration of the Polish transport sector within international initiatives;
- Analysis of the effectiveness of various forms of information and educational campaigns in changing social attitudes towards sustainable transport.

**Author Contributions:** Conceptualization, Ł.B.; methodology, Ł.B. and A.K.; validation, Ł.B.; formal analysis, Ł.B.; investigation, A.K.; data curation, Ł.B.; writing—original draft preparation, Ł.B. and A.K.; writing—review and editing, Ł.B. and A.K.; visualization, Ł.B. All authors have read and agreed to the published version of the manuscript.

**Funding:** This research received no external funding.

**Informed Consent Statement:** Informed consent was obtained from all subjects involved in the study.

**Data Availability Statement:** The data presented in this study are available on request from the corresponding author.

**Conflicts of Interest:** The authors declare no conflicts of interest.

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
