# Peer review of "Challenges of the Green Transformation of Transport in Poland"

_sustainability, doi:10.3390/su16083418_

Round 1
Reviewer 1 Report
Comments and Suggestions for Authors
In my opionon, the topic as well as the content of paper is very important from the practical perspecive, including valuable final recommendations listed within the research conclusions. Despite the paper is well structured and the Authors obtained the main aim of the study related to the challanges identification of the green transformation of transport in Poland, its academic contribution is rather average and the paper can be accepted only on condition the minor revision concerning part of desk research results.
There are two basic weaknesses of the study. First, the literature review in section 2 is limited to the wide presentation of domestic strategies, European documents (eg. Fit For 55, Green Deal, etc.) and world institutional initiatives. Therefore, in order to identify the real research gap and increse potential academic contribution, it must be also contextualized with respect to present theorethical background and previous empirical research based on systemic scientific literature review. The second weak point is the low level of innovativeness in terms of methodological approach. In this case, the Authors are fully aware that traditional research SWOT methodology based on the interviews with not representative group of 51 selected experts is the main limitations of the study and it does not require the major revision ( see lines 869-871).
Author Response
We would like to sincerely thank you for your time and effort in reviewing our article. We are grateful for the comments sent, they are extremely valuable and will allow us to develop our scientific skills and improve the article.
The article has been changed according to your suggestions, see lines:
- 100-137;
- 328-417;
- 646-761;
- 872-876;
- 877-894;
- 936-960;
- 965-984;
- 988-1004;
- 1008-1027;
- 1144-1209;
- 1277-1353;
- 1365-1375.
Reviewer 2 Report
Comments and Suggestions for Authors
I read the manuscript entitled ‘Challenges of the Green Transformation of Transport in Poland’ with interest. This paper comprehensively explores various aspects of the green transport transition in Poland, provides a comprehensive research perspective, and offers actionable and strategic recommendations that provide valuable references for addressing environmental challenges. However, I believe that the paper leaves something to be desired in terms of innovation and needs to be supplemented for better reader acceptance.
1.There is a slight gap in the clarity of Figure 3 compared to the other images, please correct this.
2.the paper provides some theoretical ideas in describing the challenges and opportunities of the green transport transition in Poland, but lacks empirical data or cases to support these ideas.
3.the paper mentions the importance of the need to invest in environmentally friendly transport infrastructure, but does not provide insights into possible technological innovations or solutions. Future research could focus on analysing the potential impacts and applications of emerging technologies (e.g. electric vehicles, renewable energy, etc.) to facilitate the practical implementation of green transport transformation.
4.It is suggested that the paper suggests some directions for future research to fill in the gaps of current research.
5.Consider more detailed analyses of different stakeholders, different regions or different transport modes.
6.Ensure that the participating experts represent various fields and stakeholders to obtain a more comprehensive and objective view.
Comments on the Quality of English LanguageModerate editing of English language required.
Author Response
We would like to sincerely thank you for your time and effort in reviewing our article. We are grateful for the comments sent, they are extremely valuable and will allow us to develop our scientific skills and improve the article.
Corrections have been made to individual comments:
1.There is a slight gap in the clarity of Figure 3 compared to the other images, please correct this.
Figure has been improved.See lines: 73-84.
2.the paper provides some theoretical ideas in describing the challenges and opportunities of the green transport transition in Poland, but lacks empirical data or cases to support these ideas.
Corrected as per comment, see lines: 73-84.
3.the paper mentions the importance of the need to invest in environmentally friendly transport infrastructure, but does not provide insights into possible technological innovations or solutions. Future research could focus on analysing the potential impacts and applications of emerging technologies (e.g. electric vehicles, renewable energy, etc.) to facilitate the practical implementation of green transport transformation.
Corrected as per comment, see lines: 646-761; 1365-1375.
4.It is suggested that the paper suggests some directions for future research to fill in the gaps of current research.
Corrected as per comment, see lines: 1365-1375.
5.Consider more detailed analyses of different stakeholders, different regions or different transport modes.
Corrected as per comment, see lines: 877-894.
6.Ensure that the participating experts represent various fields and stakeholders to obtain a more comprehensive and objective view.
Corrected as per comment, see lines: 877-894.
Reviewer 3 Report
Comments and Suggestions for Authors
The paper aims to identify and analyze the challenges of the green transformation of transport in Poland using desk research, focus interview - expert research, original approach to the use of SWOT analysis. It is interesting and well structured. However, I have some major concerns about the paper before recommendation for publication.
1. Reading your introduction, I know you want to promote Green Transformation of Transport in Poland. However, the paper is a research paper. It is important for the authors to explain their contributions to the literature. In the revisions, please clearly
2. Besides UC and Poland, there are many countries having experience in promoting green transformation of transport. Review some studies outside UC in the literature review.
3. How these 51 experts are selected? What is your selection criteria? This information is important for judge whether we can obtain scientific results from their rates.
4. Three scenarios are proposed in the section of discussion by the Polish Organization of Petroleum Industry and Trade. What is the role of this paper or authors?
Author Response
We would like to sincerely thank you for your time and effort in reviewing our article. We are grateful for the comments sent, they are extremely valuable and will allow us to develop our scientific skills and improve the article.
Corrections have been made to individual comments:
- Reading your introduction, I know you want to promote Green Transformation of Transport in Poland. However, the paper is a research paper. It is important for the authors to explain their contributions to the literature. In the revisions, please clearly
Corrected as per comment, see lines: 73-84.
- Besides UC and Poland, there are many countries having experience in promoting green transformation of transport. Review some studies outside UC in the literature review.
Corrected as per comment, see lines: 100-137; 328-417; 646-761; 1145-1209
- How these 51 experts are selected? What is your selection criteria? This information is important for judge whether we can obtain scientific results from their rates.
Corrected as per comment, see lines: 877-894.
- Three scenarios are proposed in the section of discussion by the Polish Organization of Petroleum Industry and Trade. What is the role of this paper or authors?
The study is one of the few attempts (currently available in the literature) that contains forecasts relating to possible scenarios of the green transformation of transport. In the context of the analyzes conducted in the article, it is a valuable publication, especially in the discussion of the results.
Reviewer 4 Report
Comments and Suggestions for Authors
Dear authors, thank you for sending your manuscript.
The manuscript addresses a topical issue. Unfortunately, however, the manuscript faces many problems in the areas of literature review, methodology, results, discussion, and conclusions. The manuscript does not provide an extension of scientific knowledge worthy of the global scientific community. Below is a summary of my remarks.
The manuscript provides a comprehensive overview of the challenges associated with the green transformation of transport in Poland. However, it falls short of offering new insights or significant contributions to the existing body of knowledge. The literature review and the discussion predominantly reiterate well-known facts without proposing innovative solutions or perspectives.
The methodology section lacks clarity and depth. Specifically, the description of the SWOT analysis and the selection of experts for the study do not provide enough detail to assess the robustness of the research design. Furthermore, the rationale behind the choice of experts and the statistical treatment of data need to be more explicitly articulated to enhance the study's credibility.
The analysis presented in the manuscript appears superficial, with a tendency to list factors without delving into their interconnections or the broader implications for policy and practice. A more critical and in-depth examination of the identified strengths, weaknesses, opportunities, and threats is necessary to provide valuable insights into the green transformation of transport in Poland. I am disappointed that the authors did not use a standardized SWOT analysis but adapted it to their needs (line 75).
Although the manuscript aims to identify and analyze challenges, it stops short of offering concrete recommendations or strategies for overcoming these challenges. The conclusion could be significantly strengthened by including actionable suggestions for policymakers, industry stakeholders, and the research community.
The manuscript suffers from several grammatical and typographical errors, which detract from its readability and academic rigor. A thorough proofreading and revision for language and style are required. There must always be some text between a chapter and a subchapter (2. and 2.1. for example).
Some references seem outdated or tangentially related to the manuscript's core themes. An update of the literature review to include more recent studies and relevant research would improve the manuscript's relevance and depth. The list of references is not formatted as required by the journal.
The limits of the study are not placed at the end of the discussion. The limits of the study should be more extensive and detailed. The discussion is underdeveloped and could be expanded in the context of comparison with other studies. I recommend expanding future research steps in the conclusion.
The whole manuscript is rather a case study of local (national) importance suitable as a conference paper, not for a respected impact journal.
Comments on the Quality of English LanguageModerate editing of English language required - the manuscript suffers from several grammatical errors.
Author Response
We would like to sincerely thank you for your time and effort in reviewing our article. We are grateful for the comments sent, they are extremely valuable and will allow us to develop our scientific skills and improve the article.
The article has been changed according to your suggestions:
- The introduction and literature review have been expanded in relation to the analyzed topics, see lines: 100-137;328-417; 646-761;
- The description of the methodological assumptions has been expanded, see lines: 872-876;877-894;
- Analyzes of research results have been supplemented, see lines: 936-960; 965-984; 988-1004; 1008-1027;
- The discussion of the results has been supplemented, see lines: 1144-1209;
- The conclusions of the results has been supplemented, see lines: 1277-1353; 1365-1375.
Round 2
Reviewer 3 Report
Comments and Suggestions for Authors
accepted
Reviewer 4 Report
Comments and Suggestions for Authors
Dear authors of the manuscript. Thank you for incorporating my comments. You have incorporated some of them precisely, but others less so. My main point remains, and that is the whole manuscript is rather a case study of local (national) importance suitable as a conference paper, not for a respected impact journal. However, it is not possible to incorporate this comment simply by revising the manuscript. For these reasons, I leave it to the editor to decide the future fate of your manuscript.
Comments on the Quality of English LanguageModerate editing of English language required. The manuscript still contains typos. I recommend a grammar check.